# Multi-resource dynamic coordinated planning of flexible distribution network

Rui Wang[1,6], Haoran Ji[1,6], Peng Li [1] ✉, Hao Yu [1] ✉, Jinli Zhao[1], Liang Zhao[2], Yue Zhou[3], Jianzhong Wu [3], Linquan Bai[4], Jinyue Yan [5] & Chengshan Wang[1]

The flexible distribution network presents a promising architecture to accommodate highly integrated distributed generators and increasing loads in an efficient and cost-effective way. The distribution network is characterised by flexible interconnections and expansions based on soft open points, which enables it to dispatch power flow over the entire system with enhanced controllability and compatibility. Herein, we propose a multi-resource dynamic coordinated planning method of flexible distribution network that allows allocation strategies to be determined over a long-term planning period. Additionally, we establish a probabilistic framework to address source-load uncertainties, which mitigates the security risks of voltage violations and line overloads. A practical distribution network is adopted for flexible upgrading based on soft open points, and its cost benefits are evaluated and compared with that of traditional planning approaches. By adjusting the acceptable violation probability in chance constraints, a trade-off between investment efficiency and operational security can be realised.

A distribution network serves as a critical infrastructure that delivers electricity directly to customers in a power system[1]. Owing to the low-carbon transformation in energy field[2], the distribution network is developing into a public platform that fulfils diversified user demand and enables clean energy generation[3]. Distribution network planning aims to satisfy the development of sources and loads while ensuring system security over a period by allocating various resources effectively and economically[4]. In recent years, conventional distribution networks have been inundated with significant challenges[5]. For example, renewable energy sources such as distributed photovoltaics (PVs) are widely integrated into distribution networks[6], and electric vehicles (EVs) are developing rapidly as an emerging load demand[7]. Considering China's statistics as an example, the cumulative installed capacity of distributed PVs increased by 46.61% year-on-year to 157.62 million kW in 2022[8], and the number of new energy vehicles reached 13.1F million, with a year-on-year increase of 67.13%[9]. The radial structure of a conventional distribution network renders it

difficult to manage the bidirectional power flow caused by large-scale distributed generators (DGs) and increased loads[10]. In addition, the inherent volatility originating from the sources and loads inevitably results in voltage violations, line overloads and other security issues in distribution networks[11]. Nevertheless, traditional planning approaches, such as constructing new substations and expanding feeder capacities[12], have low asset utilisation because of their insufficient flexibility and are becoming increasingly unaffordable for implementations in a well-developed urban grid. Therefore, a novel architecture for distribution networks and the corresponding planning methodology are required.

As a typical flexible distribution device, the soft open point (SOP) offers multiple advantages, such as spatial power flow regulation and real-time responses to variations[13]. Based on SOPs, a flexible distribution network (FDN) has been established, which presents a promising architecture to accommodate diverse elements and address the source-load uncertainties in a more efficient and cost-effective

[1]Key Laboratory of Smart Grid of Ministry of Education, Tianjin University, Tianjin 300072, China. [2]State Grid Tianjin Electric Power Company, Tianjin 300010, China. [3]School of Engineering, Cardiff University, Cardiff CF24 3AA, UK. [4]Department of Electrical Engineering and Computer Science, University of Tennessee, Knoxville 37996, USA. [5]School of Sustainable Development of Society and Technology, Mälardalen University, Västerås 721 23, Sweden. [6]These authors contributed equally: Rui Wang, Haoran Ji. ✉e-mail: lip@tju.edu.cn; tjuyh@tju.edu.cn

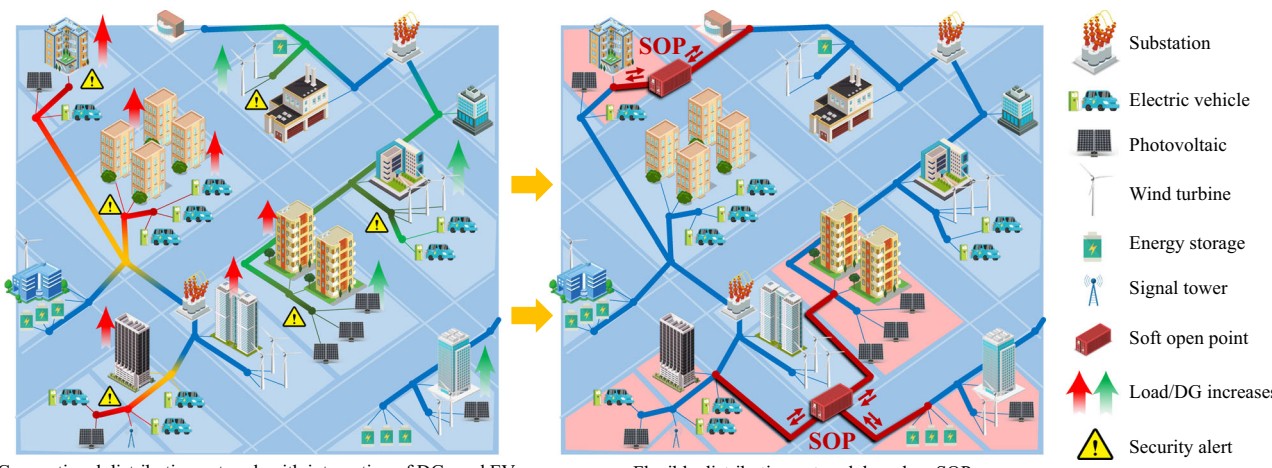

Conventional distribution network with integration of DGs and EVs | Flexible distribution network based on SOPs

**Fig. 1 | Illustration of conventional and flexible distribution networks with highly integrated DGs and EVs.** The conventional distribution network used to be built in a radial structure, especially with multi-sectioned or double-loop enhancement in urban grids. This structure is generally adopted for unidirectional power flow from generators to loads. When DGs and loads increase as time progresses, the conventional distribution network is threatened by security risks, such as voltage violations and line overloads. However, in FDNs, flexible interconnections between feeders are established based on SOPs. The sources and loads in different areas are better coordinated, and the stochastic power flow of the entire system is regulated more efficiently. Owing to its ability to accommodate large-scale DGs and EVs, the meshed architecture of an FDN offers the potential to establish an eco-friendly power supply system in an economical and efficient manner.

manner[14]. In FDNs, feeders do not operate in isolation; however, feeders that have complementary resources or suffer from violations can be interconnected flexibly[15]. With the powerful regulation of SOPs, an FDN can operate in closed loop and exchange energy across regions[16]. In addition, the common DC bus of SOP can serve as an interface for future expansion, which enables the topological evolution of FDN to satisfy the growth of sources and loads[17]. Therefore, in contrast to conventional distribution networks, the FDN exhibits a meshed architecture characterised by flexible interconnections and expansions based on SOPs, thereby providing enhanced controllability and compatibility for emerging demands. The potential of FDNs to balance energy generation and consumption between areas is exploited, and the random fluctuations in FDNs can be better alleviated, as illustrated in Fig. 1.

The grid updating and the development of DG facilities and EV charging stations (EVCSs) require a considerable amount of time, lasting for years or even decades. As a result, FDN planning entails a phased approach for reinforcement of the distribution network[18]. The terminal number and converter capacity of SOPs can expand by stage[19], which enables the configuration of SOPs to adapt to demand changes. The investment cost can be reasonably assigned to each planning stage, ensuring the consistency of FDN planning and avoiding investment reset. Hence, the dynamic evolution of FDN topology needs to be considered when decisions are required for long-term planning.

The FDN planning is conducted from the perspective of power companies, who aim to optimize the siting and sizing of SOPs for the distribution network to improve its hosting capability of EVCSs and PVs. The increasing EVCSs and PVs will change the power flow of distribution networks, which affects the location and capacity of SOPs[20]. Thus, the power companies have the motivation to perform a coordinated planning of SOPs, EVCSs, and PVs, and the FDN planning model should accommodate the allocation strategy of EVCSs and PVs. On the other hand, as the EVCSs and PVs in distribution networks are invested and built by public stakeholders, the planning schemes of EVCSs and PVs are provided as guidance and suggestions for them. Therefore, the coordinated planning of SOPs, EVCSs and PVs to maximise the overall social benefits is investigated in this paper, which provides comprehensive planning guidance for power companies,

energy suppliers and users in distribution networks. To ensure that the FDN planning strategy satisfies the security requirements in actual operations, the planning method also needs to incorporate the uncertainties of the sources and loads[21].

The motivation behind this work is to explore and design an architecture of distribution networks based on SOPs with the integration of high penetration of DGs and flexible loads. The paper highlights the flexible regulation and interconnection capabilities of SOPs in spatial dimension, which enables an interconnected and extensible architecture for distribution networks. The flexible upgrading of the distribution network can enhance its energy management and DG hosting capability[22], making it a more cost-effective alternative to constructing new substations or feeders. Therefore, the specific questions that we aim to address are, how to develop a successive FDN planning strategy over a long duration, and how to determine the siting and sizing of SOPs, EVCSs and PVs simultaneously, while considering source-load uncertainties.

The contributions of this paper are summarised as follows.

(1) A multi-resource dynamic planning method of FDNs is proposed, in which the configuration of SOPs, PVs and EVCSs is coordinated over a long-term planning period. The flexible reinforcement of the FDN can be implemented in multiple stages, and favourable cost benefits can be achieved compared with the traditional planning approach.

(2) In the FDN planning model, a probabilistic framework is established to address the strong source-load uncertainties. The security risks are formulated by chance constraints, and the stochastic nonlinear optimisation model is effectively solved based on the modified iterative algorithm. By adjusting the acceptable violation probability in chance constraints, a trade-off between investment efficiency and operational security can be obtained.

## Results
### A probabilistic framework for FDN planning
To address the uncertainties stemming from the sources and loads in FDN planning, a probabilistic framework is established, as shown in Fig. 2. We classify the framework into five main parts; FDN modelling, chance-constrained programming, uncertainty quantification, uncertainty propagation, and modified iterative algorithm.

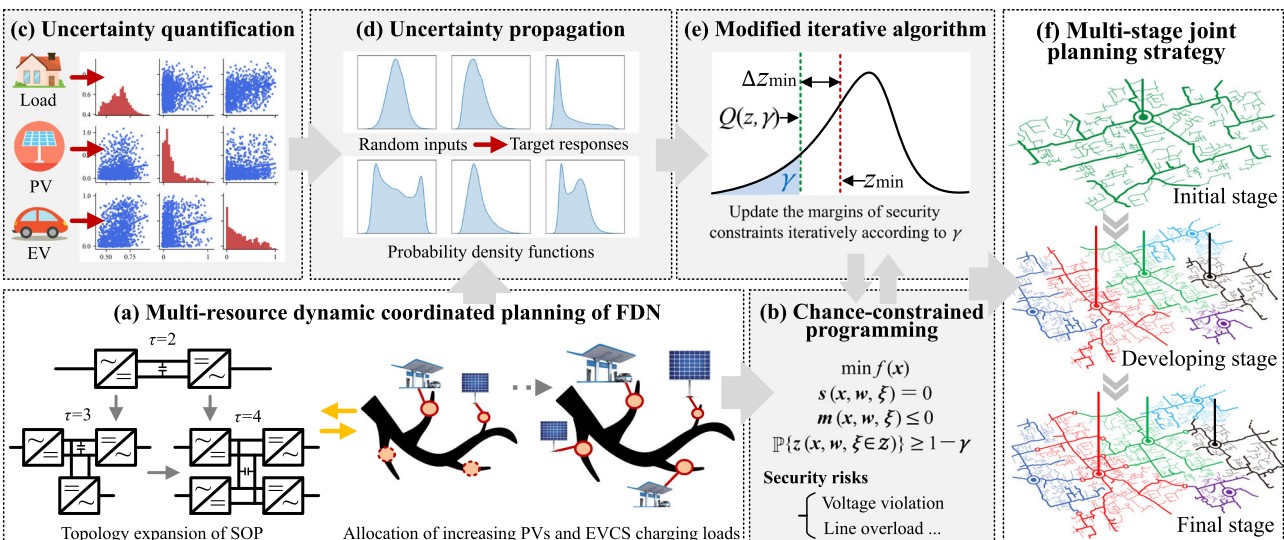

**Fig. 2 | Probabilistic framework for FDN planning. a** A multi-resource dynamic coordinated planning model for FDNs is established, where the topology evolution of SOP and the coordination of PV and EVCS are considered. **b** The security risks of FDN, including voltage violations and line overloads, are formulated by chance constraints. **c** The source-load uncertainties in FDNs are quantified based on Gaussian mixture model without relying on the assumption of typical probability functions, and the correlations between uncertainties are addressed by Nataf transformation. **d** The uncertainty propagation in FDNs can be realised based on Monte Carlo simulation, and the low-rank approximation method can be adopted as an alternative to improve computational efficiency. **e** The stochastic optimisation model is solved by the modified iterative algorithm. **f** The planning strategy for FDN is formulated.

FDN modelling is performed to mathematically describe the operation and planning mechanisms of FDN. Security risks are formulated by chance constraints. Compared with the multi-scenario analysis[23] and robust optimization method[24], chance-constrained programming can probabilistically characterise the uncertainties in FDNs, and allow a specified degree of constraint violations[25], thus enabling a trade-off between investment costs and operational security. Next, uncertainty quantification is performed to model the random variables as inputs involved in the chance constraints[26]. Uncertainty propagation is applied to obtain the statistical characteristics of FDN states, such as nodal voltages and branch currents[27], which are generally subjected to security guidelines.

Additionally, a modified iterative algorithm is developed to solve the chance-constrained FDN planning model. In this algorithm, the deterministic model with iterative formats is established by correcting the margins of security constraints based on the acceptable violation probability, which can be solved efficiently by commercial solvers in each iteration. Margin corrections are obtained via a sampling approach[28] that can manage arbitrary random inputs without the assumption of symmetric distributions.

**Practical distribution network**
A modified practical distribution network[29] is adopted to verify the effectiveness of FDN planning. The case includes 11 feeders of 11.4 kV, and the structural diagram is illustrated in Supplementary Fig. 1. The planning period is 20 years and is divided into four stages. According to the annual growth rate of conventional load, and EV and PV penetrations for each stage, the allocation of SOPs, EVCSs and PVs can be determined to accommodate the expected capacity of the sources and loads.

Price is a key factor that affects planning decisions. The prices associated with equipment investment, land exploitation, line construction, and electricity purchase are listed in Supplementary Table 1. The prices of land exploitation and line construction are assumed to increase over the planning horizon since the available urban space is gradually occupied. However, the prices of equipment and electricity are expected to decrease as manufacturing and technology advance further.

The maximum capacities of the candidate nodes for EVCS and PV are 2 MVA and 3 MVA, respectively. The power factor of the PV converter is 0.95. Considering that the efficiency of converter has reached > 98%[30], the loss factor of SOP converter is set to 0.02[31]. The maximum converter capacity of a single SOP is 10 MVA, and the unit module capacity is set to 10 kVA. The lower and upper limits of nodal voltage are set as 0.95 and 1.05 p.u., respectively. The rated current of the distribution network is 400 A, with a maximum load rate of 1.0. The maximum number of iterations $k_{max} = 30$, the sample size $N = 20{,}000$, and the acceptable violation probability $\gamma = 5\%$.

To obtain the planning results of the practical distribution network, the proposed planning method is applied, which is formally specified in the Methods section. Programs are executed using Python 3.7 on a computer with an Intel Core i7-9700 3 GHz CPU and 64 GB RAM. In addition, a cost-benefit analysis is conducted and compared with that of traditional planning approaches, where the results demonstrate that flexible upgrading offers greater economic advantages. To elucidate the efficiency of the iterative algorithm, the convergence of security risks in FDNs to a predefined level is analysed. Finally, the effect of the acceptable violation probability on planning results is investigated.

**Planning strategy formulation**
With the consideration of source-load uncertainties, five cases are designed for FDN planning, and their planning results and cost benefits are further analysed.

Case I: The multi-resource dynamic coordinated planning of FDN is performed.

Case II: The coordinated FDN planning is performed without stage division.

Case III: The energy storage systems (ESSs) planning is performed in the distribution network.

Case IV: The traditional planning method is performed, where the larger-capacity lines and transformers are invested for overloaded feeders.

Case V: The distribution network is not reinforced with the increase of sources and loads.

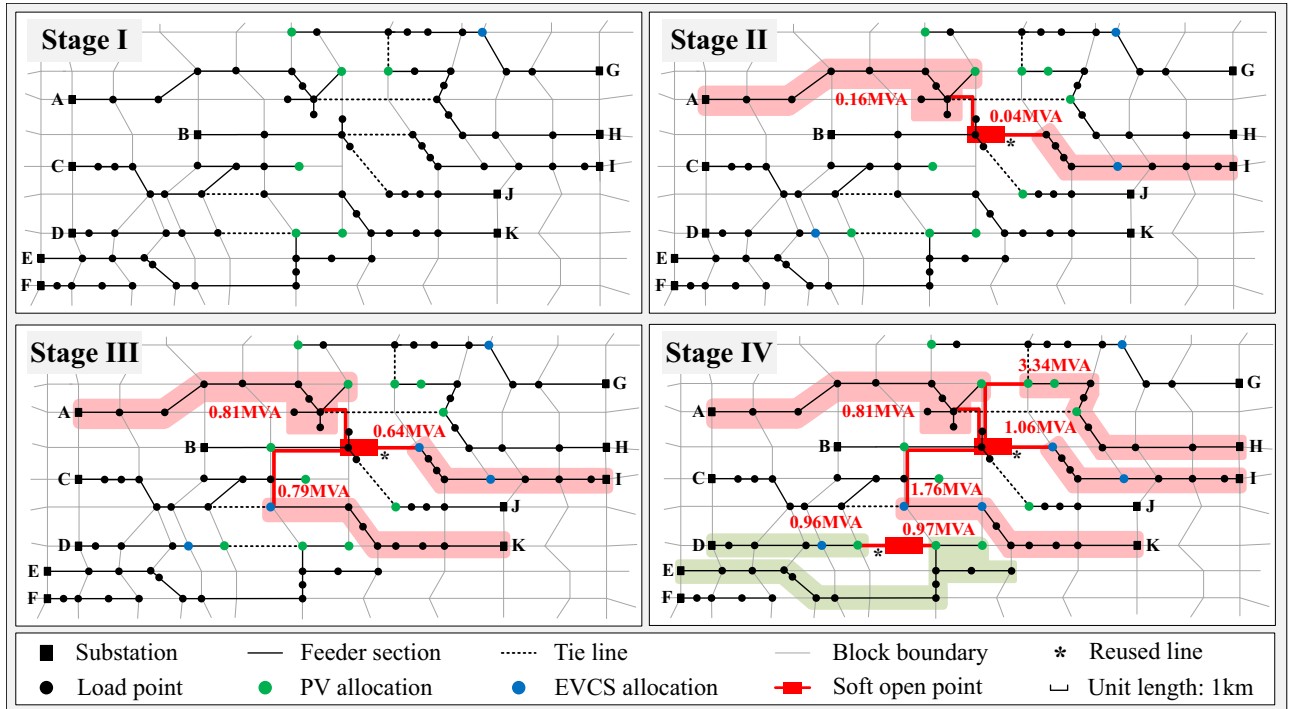

**Fig. 3 | Multi-resource dynamic coordinated planning scheme of FDN.** The annual growth rates of conventional load are 2%, 1.5%, 1% and 0.5% in Stage I-IV, respectively. EV penetration is set to 5%, 15%, 20% and 24%, and that of PV is set to 15%, 30%, 50% and 65% in Stage I-IV, respectively. Driven by the increase of sources and loads, the radial structure of the original distribution network evolved into a flexible interconnected form based on SOPs. Blue and green dots respectively denote the siting of EVCSs and PVs. The red annotations denote SOP converter capacities. Based on the coordinated planning of EVCSs and PVs, a four-terminal SOP (connecting Feeders A, H, I and K) and a two-terminal SOP (connecting Feeders D and E) are established in the final stage.

**Table 1 | Planning results of Case I**

| Stage | SOP allocation position (capacity /MVA) | EVCS allocation position (capacity /MVA) | PV allocation position (capacity /MVA) |
|---|---|---|---|
| I | - | 49(1.65) | 10(2.87) 24(0.45) 39(0.79) 40(0.04) 55(0.79) 64(0.31) |
| II | 7-72(0.16, 0.04) | 28(2.00) 49(2.00) 68(1.96) | 10(3.00) 24(1.63) 29(1.92) 39(1.81) 40(0.04) 55(2.54) 60(0.15) 63(0.70) 64(0.55) 76(0.23) |
| III | 7-72-83(0.81, 0.64, 0.79) | 28(2.00) 49(2.00) 68(2.00) 72(1.60) 83(1.27) | 10(3.00) 11(3.00) 24(2.78) 29(3.00) 39(2.81) 40(0.04) 55(3.00) 60(1.52) 63(0.74) 64(0.58) 76(2.88) |
| IV | 7-64-72-83(0.81, 3.34, 1.06, 1.76) 29-39(0.96, 0.97) | 28(2.00) 49(2.00) 68(2.00) 72(2.00) 82(1.48) 83(2.00) | 10(3.00) 11(3.00) 24(3.00) 29(3.00) 39(3.00) 40(2.72) 55(3.00) 60(3.00) 63(3.00) 64(3.00) 76(3.00) |

For Case I, the evolution of FDN throughout the planning period is shown in Fig. 3, and the siting and sizing of SOPs, EVCSs and PVs are given in Table 1. It can be observed from the results when PV and EV penetrations are low in the initial stage, no SOP is planned and the distribution network retains its original structure. In particular, EVCSs are preferentially allocated to a few positions owing to the land exploitation costs, whereas PVs exhibit a more decentralised profile to avoid over-centralization that could lead to violations. In the subsequent stages, the adoption of PVs and EVs further increases, resulting in more investment in purchasing converters and constructing charging stations.

In Stage II, a two-terminal SOP is planned owing to the heavy loads on Feeders A and I. The converter capacity of SOP in this stage is primarily used to provide reactive power compensation. Power transmission lines of the SOP are built along geographical boundaries, among which the existing switch line (12, 72) is directly reused without incurring new construction costs. In Stage III, the SOP evolves into three-terminal, enabling flexible interconnection among Feeders A, I, and K. The converter capacity of each SOP terminal also increases to transfer more active power between the feeders. In Stage IV, the original SOP develops into a four-terminal structure owing to the newly-installed PVs in Feeder H. In addition, a new two-terminal SOP is constructed between Feeders D and E, with the existing switch (29, 39) reused directly. Subsequently, the excess power generated by PVs is transferred to other feeders, ensuring that the system achieves 100% localised PV hosting within the predefined risk level.

Driven by the large-scale integration of PVs and EVs, the conventional distribution network with a multi-sectioned structure has gradually evolved into a new form with flexible interconnections based on multi-terminal SOPs. Because the dynamic expansion of SOPs and coordinated planning of PVs and EVCSs are taken into account, the investment is deferred, and the utilization efficiency is improved. The present value of the annualised costs for each stage is shown in Table 2, and the total cost is $40.70 \times 10^6$ CNY.

## Cost-benefit analysis

The economy efficiency of the above cases in alleviating security risks is further investigated. The costs of different planning schemes are illustrated in Fig. 4.

In Case II, the FDN planning is conducted within one stage, where the network evolution and the gradual growth in equipment capacity are not considered. The costs are calculated at current prices, and the

**Table 2 | Planning cost of Case I**

| Stage | Investment cost (10⁶CNY) | | | | | | Operational cost (10⁶CNY) | | | Sum (10⁶CNY) |
|---|---|---|---|---|---|---|---|---|---|---|
| | **SOP** | | | **EVCS** | | **PV** | System loss | Line overload | Voltage violation | |
| | Land exploitation | Converter purchase | Line construction | Land exploitation | Converter purchase | Converter purchase | | | | |
| I | 0.00 | 0.00 | 0.00 | 3.00 | 1.32 | 4.20 | 2.36 | 0.11 | 0.66 | 11.64 |
| II | 2.17 | 0.07 | 0.15 | 4.35 | 1.61 | 2.73 | 1.37 | 1.15 | 1.12 | 14.70 |
| III | 0.00 | 0.31 | 0.35 | 3.08 | 0.45 | 1.66 | 0.84 | 2.13 | 0.15 | 8.97 |
| IV | 1.20 | 0.32 | 0.23 | 1.20 | 0.13 | 0.45 | 0.58 | 1.18 | 0.10 | 5.38 |
| Total | 4.80 | | | 15.13 | | 9.04 | 5.15 | 4.56 | 2.02 | 40.70 |

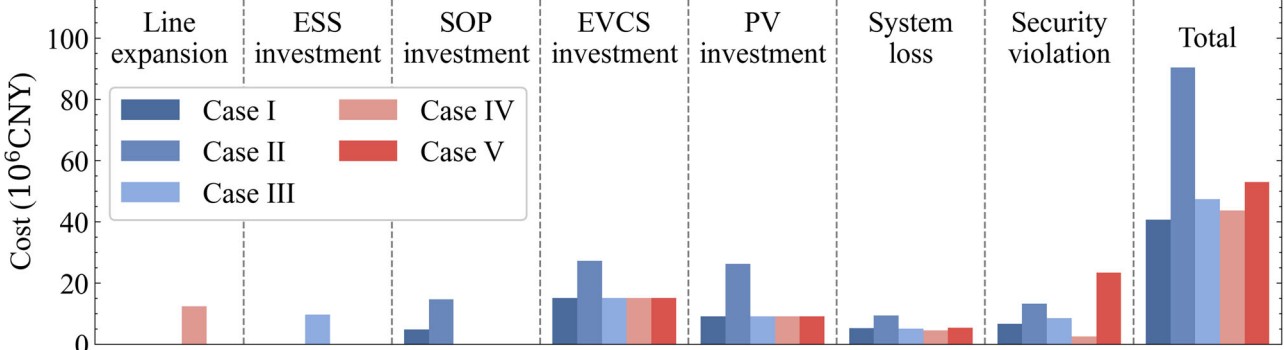

**Fig. 4 | Costs of different planning cases.** Under a moderate investment cost for establishing a flexible interconnected structure based on SOPs, the operational penalty cost for voltage violations and line overloads can be reduced. Compared with Case II, the multi-stage planning framework can delay investment. Compared with Case III, the resource sharing is promising based on the flexible interconnected structure. Compared with Case IV, the proposed planning method exhibits better flexibility for expansion. Compared with Case V, the proposed planning method effectively addresses the security issue caused by the integration of PVs and EVCSs.

investments are paid at the initial to meet the needs over the entire planning period. The total cost is the highest at 90.48 × 10⁶ CNY.

In Case III, ESSs are planned to ensure the safe operation of the distribution network. The ESS investment is used to build the site and purchase converters and batteries. The allowed minimum and maximum state of charge (SOC) of ESSs are set to 10% and 90%. Assume that the battery of ESS can be charged or discharged by its maximum available capacity, and the sequential energy constraints are not considered, which will result in a smaller capacity investment cost of the battery. However, confined to the radial structure of the distribution network, ESSs need to be installed independently at each overrunning feeder, thus causing a higher site construction cost. The total cost of planning ESSs is larger than that of planning SOPs in Case I, indicating that the flexible interconnected structure based on SOPs is economically promising for it offers resource sharing among feeders.

In Case IV, the feeders that violate security criteria in each stage are identified to be reinforced. In this case, Feeders A and I in Stage II, Feeder K in Stage III, and Feeders E and H in Stage IV are reinforced. The maximum capacity of the expanded line is 1.5 times that of the original line, and the cost of expanding the line per unit length is thrice that of line construction[32]. Additionally, the transformer capacity of the reinforced feeder needs to be expanded as well. The operational risk of the distribution network is reduced by planning larger-capacity lines and transformers. However, inadequate flexibility in capacity allocation leads to higher expansion costs. The total cost reaches 43.63 × 10⁶ CNY.

Planning different resources to enhance the flexibility of the distribution network may incur additional investment expenses; however, it can reduce operational costs, such as those associated with system losses, voltage violations, and line overloads. The results of Case V show that not implementing planning measures saves the investment cost, but voltage violations and line overloads severely jeopardise the

safe operation of the distribution network, thus resulting in a higher penalty cost.

In summary, a dynamic coordinated FDN planning method is adopted in Case I, which alleviates operational risks while maintaining a lower investment cost. The total cost reduces by 55.02%, 14.06%, 6.72%, and 23.08% as compared with those of Case II-V, respectively. The results indicate that the flexible upgrading based on SOPs in a multi-stage framework offers better economy efficiency for managing security risks in distribution networks.

### Probabilistic analysis in iterative solution

There is no analytical representation for chance constraints. Direct methods involve high-dimensional integral operations. Therefore, a modified iterative algorithm is executed to obtain an efficient solution for non-linear systems, where the predefined risk margin and computational performance are guaranteed.

First, the source-load uncertainties in FDNs are quantified. The probability density functions of conventional load, EVCS charging load and solar radiation are established using Gaussian mixture models (GMMs) based on historical observations[33–35], as shown in Supplementary Fig. 2. The probability density functions of sources and loads are not symmetric and cannot be accurately described by any typical distributions, like Normal, Beta or Weibull distribution. In addition, the dependences between uncertainties are also considered, which reflect the variation consistency of random variables and are generally defined by a correlation matrix. Specially, the randomness of the PV output is mainly owing to the volatility of solar radiation, so we primarily quantify the uncertainty in solar radiation for further calculations[36].

Based on the quantified source-load uncertainties, the probability density functions of FDN states are obtained at each iteration, which reflect operational profiles and provide statistical data for iterative corrections. In the 0-th iteration, the probability density functions of

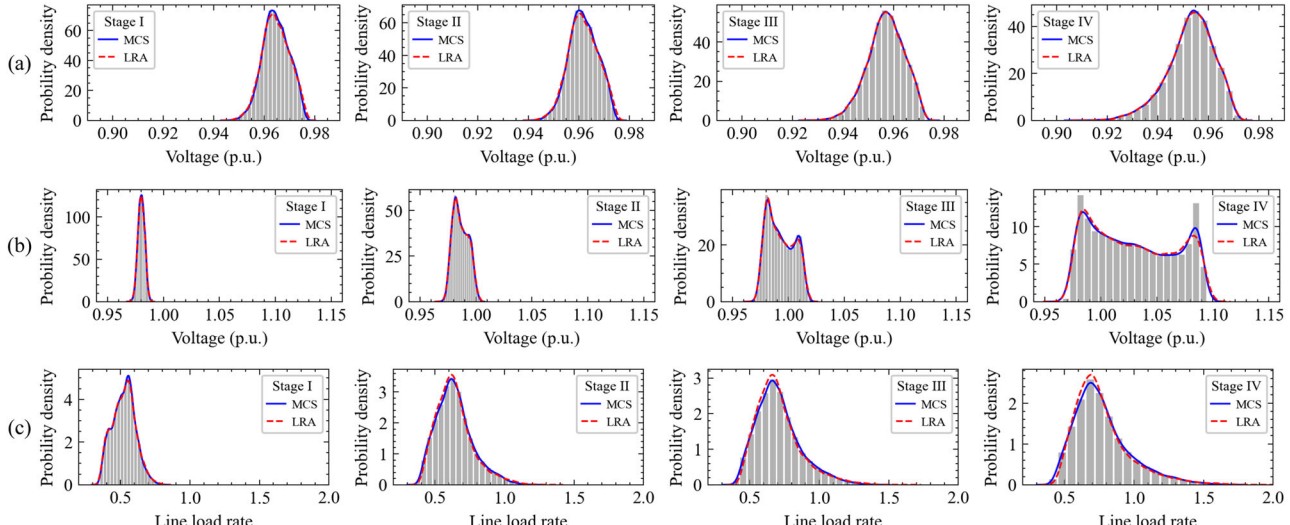

**Fig. 5 | Probability density functions of nodal voltages and line load rates.** At each iteration, an FDN planning strategy is formulated, and the operational states of FDN are obtained by uncertainty propagation. The violation probabilities of nodal voltages and line load rates are identified, then the corrected margins are adopted for solving the FDN planning model at the next iteration. Compared with Monte Carlo simulation (MCS), low-rank approximation (LRA) method can produce similar results, but with a lower computational burden. **a** Probability density functions of node 83 voltage. **b** Probability density functions of node 64 voltage. **c** Probability density functions of line (109, 65) load rate.

node 83 voltage on Feeder K, node 64 voltage on Feeder H, and line (109, 65) load rate on Feeder I over the entire planning horizon are selected for illustrations, as shown in Fig. 5. It can be found that there is an under-voltage violation risk at node 83 due to the heavy load of Feeder K, an over-voltage violation risk at node 64 due to the PV integration of Feeder H, and an overloading risk at line (109, 65). In addition, the variances of voltages and load rates progressively increase by stage, implying that the random fluctuations of FDN states are magnified and the operational profiles are exacerbated with the growing penetration of PVs and EVCSs.

To solve the stochastic optimisation for FDN planning, the modified iterative algorithm is proposed in the paper. To analyse the performance of the solution algorithm, a comparison between the general and the modified iterative algorithm is conducted. The solution results obtained by the two algorithms are almost the same, but the general algorithm necessitates 21 iterations to converge, which requires 57.46 h. However, using the modified iterative algorithm, only 6 iterations are necessitated to attain convergence, which requires 9.40 h. In each iteration, the deterministic optimisation model is solved within 49.93 min on average, and optimal solution is guaranteed with the maximum gap smaller than tolerance 1e-3. The comparison indicates that the modified iterative algorithm has a better performance, and its improvements are described in detail mathematically in the Methods section. Moreover, by using low-rank approximation to replace Monte Carlo simulation for uncertainty propagation, the convergence time can be further reduced to 5.53 h.

As shown in Fig. 6, the violation probabilities exceed 30% when no SOP is implemented at the initial iteration. By correcting the upper and lower margins of security constraints iteratively, the planning strategy is resolved and the violation risks under the solution gradually converge to a predefined level. In this case, after four iterations, the risks of voltage violations and line overloads are controlled within 5%.

### Scenario analysis

Four scenarios are designed with acceptable violation probabilities of 3%, 5%, 10%, and 15%. The dynamic planning schemes for FDN are shown in Fig. 7. When the acceptable violation probability is lower, the SOP is constructed in the earlier stage with a larger converter capacity. For example, when $\gamma$ = 3%, a two-terminal SOP is built in Stage I to address the under-voltage risk caused by the heavy load at the end of Feeder A. When $\gamma$ = 5% in Case I, SOP planning begins in Stage II. When $\gamma$ = 10% and 15%, the planning of SOP is postponed to Stage III. In the final stage, four- and three-terminal SOPs are built when $\gamma$ = 3%, and Feeders A, B, E, H, I and K are interconnected together for resource sharing and power regulations. When $\gamma$ = 5% and 10%, four- and two-terminal SOPs are planned, whereas three- and two-terminal SOPs are constructed when $\gamma$ = 15%. the total SOP converter capacities of the four scenarios are 9.54, 8.90, 7.98, and 6.35 MVA, respectively.

The cost benefits of the above scenarios are further analysed, as shown in Fig. 8. The acceptable violation probability of FDN has a significant impact on the cost. A lower permitted risk corresponds to a more conservative planning scheme, along with a higher investment cost, but a lower operational penalty cost. When $\gamma$ = 3%, the total cost of FDN planning is minimised. In practical engineering, the trade-off between economy and security can be achieved by adjusting the acceptable violation probability based on actual demands.

## Discussion

The flexibly interconnected and extensible architecture enables FDN to dispatch power flow over the entire system in closed-loop operation. This architecture is based on SOPs that provide strong controllability for wide-area active power transfer and local reactive power compensation. Consequently, the FDN offers a promising way to realize capacity expansion and low-carbon transformation in power systems with highly integrated PVs and EVs. The topology of distribution network is progressively updated and enhanced by segmenting the long planning period into several stages. The establishment of FDN will take years or even decades to fulfil the developing needs of users. A four-terminal SOP may evolve into a six- or eight-terminal structure that encompasses more power supply areas. Compared with traditional planning approaches, such as constructing new substations and expanding the capacity of feeders, the flexible evolution of FDN enables significant cost reduction.

SOP takes part as the key infrastructure for the structure evolution of distribution networks, which is the priority to be considered in the FDN planning. Other controllable resources, such as energy storage systems and demand responses, can be further considered in subsequent research. In addition, the common DC bus of SOP is a

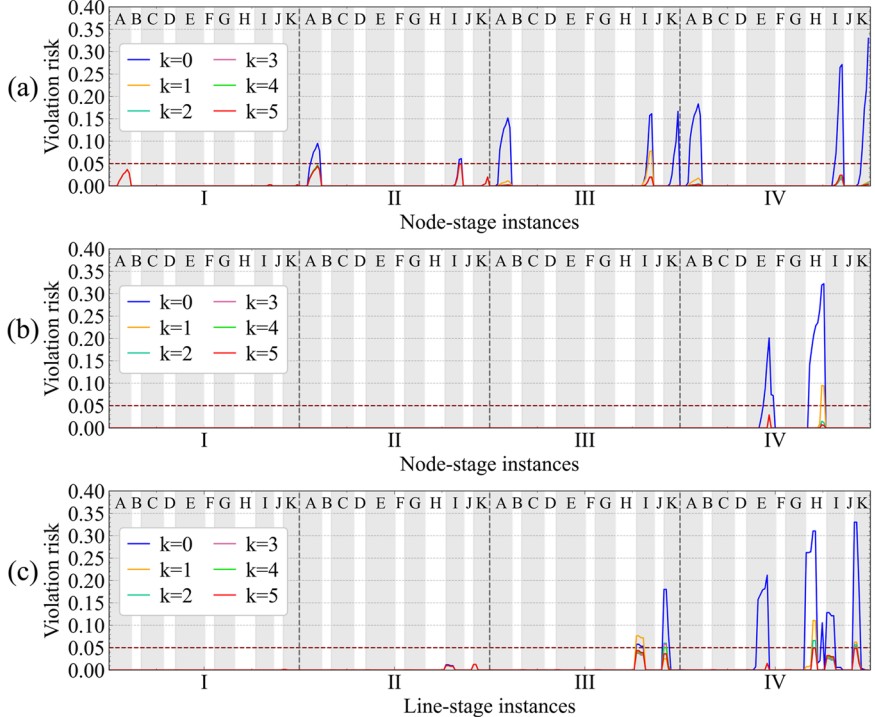

**Fig. 6 | Violation risks of FDN during iterations.** FDN planning strategy is updated iteratively with the corrected margins of security constraints, thus ensuring the violation risks converge to the predefined level. Convergence efficiency is improved using the modified iterative algorithm. The x-axis ticks represent the indices of stage. At each stage interval (divided by grey dashed lines), the violation risks of nodal voltages or line currents are illustrated, with labels (A, B, ..., K) at the top indicating corresponding feeder names. The y-axis ticks represent the violation risks. The initial violation risks (blue lines) are rapidly reduced to the vicinity of 5% (dark red dashed line) only after one iteration (orange lines). Then, slight reductions of violation risks are conducted during later iterations. The iteration stops (red lines) when all violation risks are controlled below the acceptable probability. **a** Violation risk of the lower bound for voltages. **b** Violation risk of the upper bound for voltages. **c** Violation risk of the upper bound for currents.

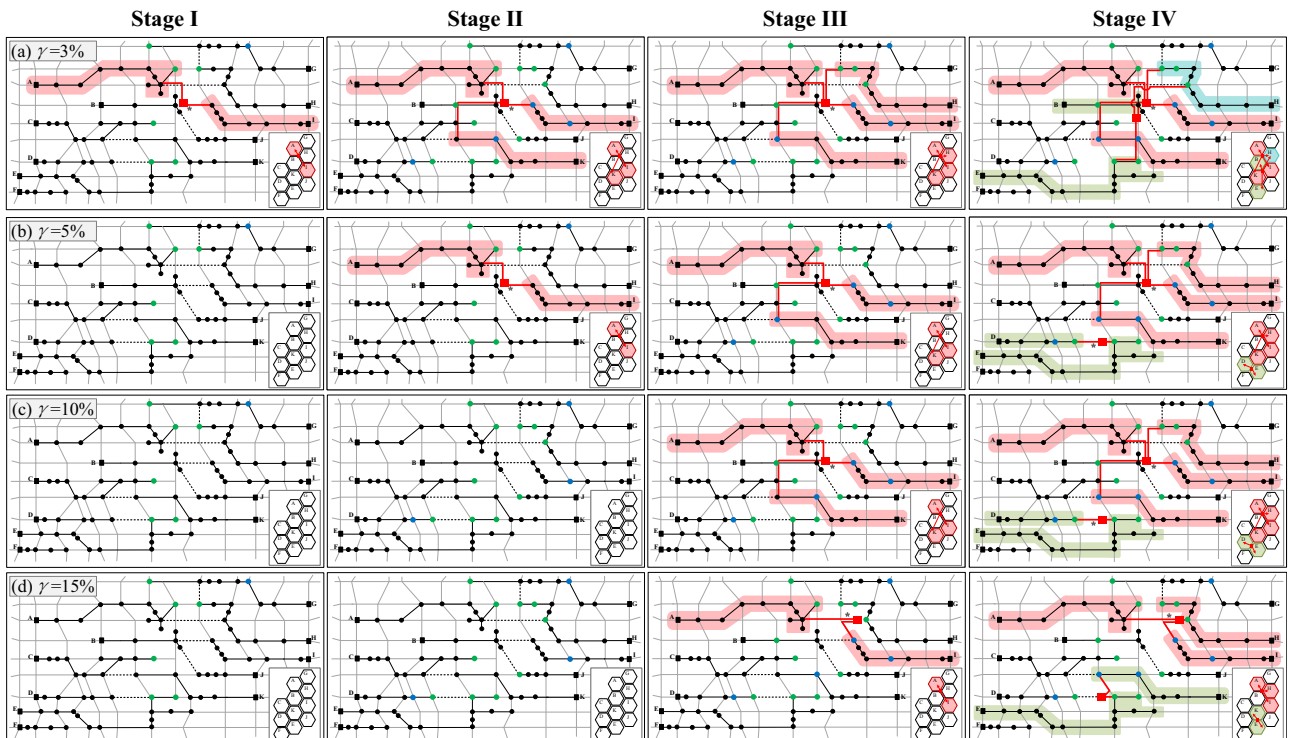

**Fig. 7 | Planning results of FDN with different acceptable violation probabilities.** The stage to begin flexible reconstruction and the final evolutionary topology of the distribution network differ in terms of the acceptable violation probabilities. Owing to adapting the same penetrations of sources and loads in each stage, the allocations of PVs and EVCSs in the four scenarios are almost the same. The feeders connected by the same SOP are represented in the same colour. When $\gamma = 3\%$, Feeder H is connected by two SOPs in Stage IV simultaneously, which is represented in an additional colour. **a** $\gamma = 3\%$. **b** $\gamma = 5\%$. **c** $\gamma = 10\%$. **d** $\gamma = 15\%$.

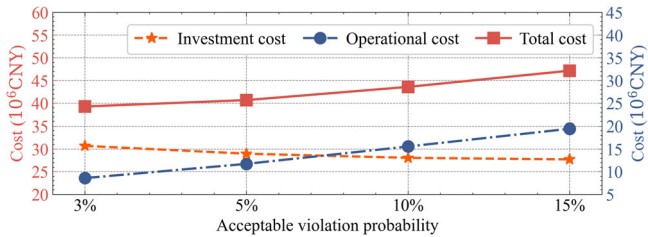

**Fig. 8 | Cost analysis of the scenarios with different acceptable violation probabilities.** Golden and red points respectively denote investment and total costs, referring to the left y-axis. Blue points denote operational cost for system loss, voltage violation, and line overload, referring to the right y-axis.

compatible interface that provides access to DC loads, DC sources, and energy storage systems. In summary, FDN is both eco-friendly and economical to host emerging elements with diverse characteristics.

To address the high-dimensional uncertainties derived from the sources and loads in FDNs, we establish a probabilistic framework. Violation risks are formulated by chance constraints, and a modified iterative algorithm is developed to solve the chance-constrained optimisation problem. In our example, the solution can be obtained within hours via a few iterations, and the security risks are reduced to an acceptable level. In practice, the decision maker can modify the FDN planning strategy by adjusting the acceptable threshold to achieve a balance between economy and security.

With the orientation of coordinated planning results, the EVCSs and PVs constructed at the suggested positions will be better regulated and supported by SOPs to ensure the secure hosting of renewable generation and charging load. Therefore, it is beneficial to formulate planning guidance for individual users, especially for the energy suppliers that make profits by providing public generating and charging services. With the simultaneous consideration of grid, source and load in a multi-stage planning model, the planning guidance exhibits high compatibility for all participants in the distribution network. In this way, the construction of distribution network can cover the interests of multiple parties and offer better power supply service in practical operations. At the same time, note that the energy suppliers and users are not mandated to follow the allocation result of sources and loads. But the user-built EVCSs and PVs may bring operational violations to the distribution network, resulting in the inability to plug themselves into the grid. Under such conditions, users have to pay additional costs to dispatch energy storage systems or other controllable resources for regulations, while wasting the public services provided by configured SOPs.

An additional case is investigated as Case VI, where the siting and sizing of SOPs are formulated without the consideration of increasing EVCSs and PVs, while the allocation of EVCSs and PVs is determined and invested by users. Compared with Case I, the number and capacity of SOP in Stage III-IV are smaller, and SOP investment cost is reduced by 9.17%. However, for a stochastically generated allocation of EVCSs and PVs, the voltage violation risk of the distribution network in Stage II-IV exceeds 15%, while the line overload risk exceeds 25%. As a result, the operational cost for violations increases by 4.70 times, and the total cost increases by 3.31 times. The case indicates that if FDN planning is not performed in a coordinated framework, the regulation ability of SOPs cannot be fully utilized, and the randomness caused by sources and loads will exacerbate the operation of distribution network. The separate decision-making leads to worse economic outcomes for the overall benefit of society.

The coordinated planning method of FDN is essentially a multi-objective optimization model. In this paper, the multiple interests are formulated and normalized as costs, which can be summed as a single-objective function in a straightforward way. Then, the FDN planning

model can be converted into a mixed-integer second-order conic programming (MISOCP) problem, which can be solved effectively by commercial solvers. In this sense, the proposed planning method is to maximise social benefits and achieve the overall economy efficiency for power companies, energy suppliers, and users. As for multi-objective optimization problems, the Pareto frontier is also known as a promising approach, which provides a wide range of alternative solutions for decision-making. In the Pareto frontier, each objective is considered as equally good. Additionally, game theory is also suited to solving multi-stakeholder problems based on Nash equilibrium, where the interests of power companies, energy suppliers, and users are optimized simultaneously. This paper focuses on how to establish a flexible interconnected architecture of FDN based on SOPs, and the handling of multi-objective optimization will be further studied in future works due to the limited space.

The simultaneous use of SOP and line reinforcement is further investigated as Case VII. In a feasible planning scheme, Feeder K is reinforced in Stage III, and a two-terminal SOP is planned in Stage II and evolves into a four-terminal structure in the final stage with the coordination planning with EVCSs and PVs. Compared with Case I, its investment cost ($29.72 \times 10^6$ CNY) is larger, but its operational cost ($9.26 \times 10^6$ CNY) is much smaller. As a result, the total cost of Case VI is reduced by $1.71 \times 10^6$ CNY (reduction of 4.20%), which exhibits potential economy efficiency.

Under failure conditions, the loads and DGs on the faulted feeder can be transferred via SOPs without incurring any power outages. The interconnected architecture has the potential to improve the load recovery in FDNs. However, this implies that the transfer capacity of SOP needs to be further optimised. FDN planning considering reliability enhancement will be investigated in future studies.

The paper mainly studies the method of how to establish a flexibly interconnected and extensible architecture of the distribution network based on multi-terminal SOPs, which has laid the foundation for the realisation of the honeycomb FDN[37]. As shown in Fig. 7, the power supply areas are abstractly denoted as a couple of closely packed hexagons, thus representing a primary visualisation of the honeycomb FDN. In our prospect, the honeycomb distribution system may be an advanced FDN structure in the future, which enables a more robust grid, by segmenting it into largely autonomous cells. It can be applied to the current grid step by step and may contribute to increasing the penetration of renewable energy resources. The decision maker will focus on the planning of flexible interconnections and expansions from a global perspective, whereas the sources, loads, and energy storage systems are self-organised in each local area.

We conclude that the multi-resource dynamic and coordinated planning method of FDN is feasible and advantageous. The probabilistic framework aimed at addressing source-load uncertainties effectively confines security risks within a predefined range. The case study demonstrates that the proposed iterative algorithm performs efficiently in solving chance-constrained programming problems. The flexible architecture and probabilistic planning method of FDN allow it to host high-penetration PVs and EVs in power systems.

## Methods
### Solution procedure
The detailed procedure for solving the FDN planning model is given as follows.

1) Input the distribution network parameters, and determine the random variables and correlation matrix;

2) Quantify the source-load uncertainties based on Gaussian mixture model in Eq. (1) and generate samples based on Nataf transformation in Eq. (2);

3) Set iteration counter $\kappa = 0$;

4) Check whether $\kappa$ is less than or equal to $\kappa_{max}$. If satisfied, continue to Step 5; otherwise, proceed to Step 9;

5) Solve the deterministic planning model in Eq. (25) to obtain the planning strategy, including the optimised allocation of SOPs, EVCSs and PVs;

6) Execute uncertainty propagation to obtain the probabilistic characteristics of nodal voltages and line currents;

7) Check whether all chance constraints satisfy the predefined risk level. If satisfied, proceed to Step 9; otherwise, continue to Step 8;

8) Update the bounds of the security constraints; update $\kappa = \kappa + 1$ and proceed to Step 4;

9) Record the solved planning strategy and calculate the total cost.

## Gaussian mixture model

The conventional load of residents, the charging load of EVCSs, and the output of PVs are regarded as random variables in the FDN planning model, and their probability density functions can be formulated by Gaussian mixture model as follows.

$$p(\boldsymbol{\xi}) = \sum_{m=1}^{M} \pi_m N(\boldsymbol{\xi}; \boldsymbol{\mu}_m, \boldsymbol{\Sigma}_m) \tag{1a}$$

$$\sum_{m=1}^{M} \pi_m = 1 \tag{1b}$$

$$N(\boldsymbol{\xi}; \boldsymbol{\mu}_m, \boldsymbol{\Sigma}_m) = \frac{\exp\left(-\frac{1}{2}(\boldsymbol{\xi} - \boldsymbol{\mu}_m)^{\mathrm{T}} \boldsymbol{\Sigma}_m^{-1}(\boldsymbol{\xi} - \boldsymbol{\mu}_m)\right)}{\sqrt{\det(2\pi \boldsymbol{\Sigma}_m)}} \tag{1c}$$

where $M$ denotes the maximum number of components. $\pi_m$ denotes the weighting factor. $N(\boldsymbol{\xi}; \boldsymbol{\mu}_m, \boldsymbol{\Sigma}_m)$ presents a Gaussian distribution with mean vector $\boldsymbol{\mu}_m$ and covariance matrix $\boldsymbol{\Sigma}_m$. In addition, the dependence between random variables is described by Pearson correlation matrix.

## Nataf transformation

Consider a $n$-dimensional random vector $\boldsymbol{\xi} = (\xi_1, \ldots, \xi_n)$ with correlation matrix $\boldsymbol{\rho}$, where element $\rho_{ij}$ denotes the correlation coefficient between variables $\xi_i$ and $\xi_j$. With Nataf transformation[38], a new random vector $\boldsymbol{\varsigma} = (\varsigma_1, \ldots, \varsigma_n)$ in standard normal space and its correlation matrix $\boldsymbol{\rho}^{\phi}$ can be obtained.

$$\varsigma_i = \Phi^{-1}(G_i(\xi_i)) \tag{2a}$$

$$\rho_{ij} = \iint \frac{G_i^{-1}(\Phi(\varsigma_i)) - \mu_i}{\sigma_i} \times \frac{G_j^{-1}(\Phi(\varsigma_j)) - \mu_j}{\sigma_j} \times \phi_2\left(\varsigma_i, \varsigma_j, \rho_{ij}^{\phi}\right) \mathrm{d}\varsigma_i \mathrm{d}\varsigma_j \tag{2b}$$

where $G_i(\cdot)$, $G_i^{-1}(\cdot)$, $\mu_i$ and $\sigma_i$ respectively denote the cumulative distribution, inverse cumulative distribution, mean and standard variance of $\xi_i$. $\Phi(\cdot)$ and $\Phi(\cdot)$ respectively denote the cumulative distribution and inverse cumulative distribution of univariate standard normal distribution. $\phi_2(\cdot)$ denotes the bivariate standard normal distribution. Element $\rho_{ij}^{\phi}$ denotes the correlation coefficient between variables $\varsigma_i$ and $\varsigma_j$. Furthermore, the independent random vector $\zeta$ in standard normal space can be obtained based on Choleskey decomposition.

$$\boldsymbol{\rho}^{\phi} = \boldsymbol{L}\boldsymbol{L}^{\mathrm{T}} \tag{3a}$$

$$\zeta = \boldsymbol{L}^{-1}\boldsymbol{\varsigma} \tag{3b}$$

where $\boldsymbol{L}$ denotes the lower triangular matrix.

## Low-rank approximation

Low-rank approximation is used to express the target response in highly compressed formats as the sum of rank-one functions via canonical decomposition.

Consider an independent random vector $\boldsymbol{\xi} = (\xi_1, \ldots, \xi_n)$ with marginal distribution $g_i(i = 1, \ldots, n)$, namely the probability density functions of sources and loads established based on Gaussian mixture models. The desired response $h$ of FDN, such as the nodal voltage and line current, can be formulated as follows.

$$h = f^{\mathrm{LRA}}(\boldsymbol{\xi}) = \sum_{l=1}^{r} b_l \omega_l(\boldsymbol{\xi}) = \sum_{l=1}^{r} b_l \left(\prod_{i=1}^{n} \upsilon_l^{(i)}(\xi_i)\right) \tag{4}$$

where $b_l$ denotes the weighting factor. $\upsilon_l^{(i)}$ denotes the $i$-th dimensional univariate function of the rank-one function $\omega_l$. In practice, $\upsilon_l^{(i)}$ is expanded on a polynomial basis $\left\{\chi_q^{(i)}, q \in \mathbb{N}\right\}$ in practice, which is orthogonal to the function $g_i$. Thus, the rank-$r$ approximation of $h$ results in the following form.

$$h = \sum_{l=1}^{r} b_l \left(\prod_{i=1}^{n} \left(\sum_{q=0}^{\theta} z_{q,l}^{(i)} \chi_q^{(i)}(\xi_i)\right)\right) \tag{5}$$

where $\chi_q^{(i)}$ denotes the $q$-th degree univariate polynomial in the $i$-th random variable. $z_{q,l}^{(i)}$ is the coefficient of $\chi_q^{(i)}$ in the $l$-th rank-one function, and $\theta$ is the maximum degree.

To determine the parameters in low-rank approximation, the sequential correction-updating algorithm[39] is employed. In addition, to tackle the correlation between random variables, the inverse Nataf transformation $\mathcal{T}^{-1}(\cdot)$ is introduced. The response $h$ can be expressed as $h = f(\boldsymbol{\xi}) = f(\mathcal{T}^{-1}(\zeta))$, where $\zeta$ is sampled from independent standard normal distributions.

## Evolution of FDN planning

The number of terminals, and the siting and sizing of SOP can be flexibly designed in each stage. First, a set of available nodes is determined for SOP connections. In this paper, the terminal nodes of existing tie lines are selected as the available nodes. Second, the topologies of SOP planning schemes are generated without exceeding the maximum number of SOP terminals, and the length of the line to be reconstructed in each scheme is calculated.

$$\mathcal{L} = \mathrm{U}_{k=1}^{N_f} \mathcal{L}(k) = \mathrm{U}_{k=1}^{N_f} \mathrm{U}_{\tau=2}^{M_f} \mathcal{L}(k, \tau) \tag{6a}$$

$$\mathcal{L}(k, 2) = \{k | \mathrm{crad}(\Omega_k) = 2\} \tag{6b}$$

$$\mathcal{L}(k, 3) = \left\{k' | \mathrm{crad}(\Omega_{k'}) = 3, \Omega_{k'} \supseteq \Omega_k\right\} \tag{6c}$$

$$\mathcal{L}(k, 4) = \left\{k'' | \mathrm{crad}(\Omega_{k''}) = 4, \Omega_{k''} \supseteq \Omega_k\right\} \tag{6d}$$

$$\mathcal{L}_i = \{k | \Omega_k \ni i, \forall i \in \Omega_s\} \tag{6e}$$

where $\Omega_s$ denotes the set of available nodes. $k$ denotes the scheme index. $\Omega_k$ denotes the set of nodes in scheme $k$. $N_f$ denotes the number of schemes. $M_f$ denotes the maximum number of SOP terminals. $\mathcal{L}$ denotes the set of total schemes. $\mathcal{L}_i$ denotes the set of SOP planning schemes containing node $i$. $\mathcal{L}(k)$ denotes the set of schemes evolved from scheme $k$, and $\mathcal{L}(k, \tau)$ denotes the $\tau$-terminal SOP planning schemes in set $\mathcal{L}(k)$. $\mathrm{crad}(\cdot)$ denotes the cardinality of a set.

## Objective function of FDN planning

The FDN planning model is established to minimise the comprehensive expense, including investment cost $\phi_u^{CO}$ and operational cost $\phi_u^{OP}$.

$$f = \min \sum_{u\in\Omega_U} \sum_{y\in\Omega_Y} \lambda_{yu}\left(\varepsilon\phi_u^{CO} + \phi_u^{OP}\right) \tag{7a}$$

$$\lambda_{yu} = (1+d)^{-[(u-1)Y+y]} \tag{7b}$$

$$\varepsilon = d(1+d)^L / \left[(1+d)^L - 1\right] \tag{7c}$$

where $\Omega_U$ and $\Omega_Y$ denote the set of stages and years, respectively. $Y$ denotes the duration of each planning stage. $y$ denotes the index of years in each planning stage. $\varepsilon$ denotes the capital recovery factor, which share the construction costs equally to each year of the payback period $L$. $\lambda_{yu}$ denotes the present value coefficient, which calculates the present value of an annualized cost in terms of interest rate $d$.

The investment cost primarily includes the cost for building SOPs, EVCSs, and PVs for land exploitation, converter purchases and line construction. Meanwhile, the operational cost is mainly attributed to network and SOP converter losses.

$$\phi_u^{CO} = \phi_u^{SOP,ST} + \phi_u^{SOP,CT} + \phi_u^{SOP,BR} + \phi_u^{EVCS,ST} + \phi_u^{EVCS,CT} + \phi_u^{PV,CT} \tag{8a}$$

$$\phi_u^{SOP,ST} = c_u^{ST}\left(\sum_{k\in\mathcal{L}}\alpha_{k,u} - \sum_{k\in\mathcal{L}}\alpha_{k,u-1}\right) \tag{8b}$$

$$\phi_u^{SOP,CT} = c_u^{CT}\left(\sum_{k\in\mathcal{L}}S_{k,u}^{SOP} - \sum_{k\in\mathcal{L}}S_{k,u-1}^{SOP}\right) \tag{8c}$$

$$\phi_u^{SOP,BR} = c_u^{BR}\left(\sum_{k\in\mathcal{L}}D_k\alpha_{k,u} - \sum_{k\in\mathcal{L}}D_k\alpha_{k,u-1}\right) \tag{8d}$$

$$\phi_u^{EVCS,ST} = c_u^{ST}\left(\sum_{i\in\Omega_e}\beta_{i,u} - \sum_{i\in\Omega_e}\beta_{i,u-1}\right) \tag{8e}$$

$$\phi_u^{EVCS,CT} = c_u^{CT}\left(\sum_{i\in\Omega_e}S_{i,u}^{EVCS} - \sum_{i\in\Omega_e}S_{i,u-1}^{EVCS}\right) \tag{8f}$$

$$\phi_u^{PV,CT} = c_u^{CT}\left(\sum_{i\in\Omega_g}S_{i,u}^{PV} - \sum_{i\in\Omega_g}S_{i,u-1}^{PV}\right) \tag{8g}$$

where $\phi_u^{SOP,ST}$, $\phi_u^{SOP,CT}$ and $\phi_u^{SOP,BR}$ denote the land exploitation, converter purchase and line construction cost of SOP in stage $u$, respectively. $\phi_u^{EVCS,ST}$ and $\phi_u^{EVCS,CT}$ denote the land exploitation and converter purchase cost of EVCS in stage $u$. $\phi_u^{PV,CT}$ denotes the converter purchase cost of PV in stage $u$. $c_u^{ST}$, $c_u^{CT}$ and $c_u^{BR}$ denote the price of land exploitation, converter purchase and line construction in stage $u$, respectively. $\Omega_e$ and $\Omega_g$ denote the nodes available for EVCS and PV installations, respectively. $\alpha_{k,u}$ is a binary variable, indicating whether SOP planning scheme $k$ is selected in stage $u$. $\beta_{i,u}$ is a binary variable, indicating whether the EVCS is constructed at node $i$ in stage $u$. $S_{k,u}^{SOP}$ denotes the converter capacity of SOP planning scheme $k$ in stage $u$. $S_{i,u}^{EVCS}$ and $S_{i,u}^{PV}$ denote the converter capacity of EVCS and PV at node $i$ in stage $u$, respectively. $D_k$ denotes the length of the line to be

constructed in SOP planning scheme $k$. The existing tie lines involved in scheme $k$ do not introduce new investment, which can be directly utilised.

$$\phi_u^{OP} = 8760 \cdot \left(\phi_u^{NET,LS} + \phi_u^{SOP,LS}\right) \tag{9a}$$

$$\phi_u^{NET,LS} = c_u^{SL}\sum_{ij\in\Omega_b} R_{ij}I_{ij,u}^2 \tag{9b}$$

$$\phi_u^{SOP,LS} = c_u^{SL}\sum_{k\in\mathcal{L}}\sum_{i\in\Omega_k} P_{i,k,u}^{SOP,LS} \tag{9c}$$

where $\phi_u^{NET,LS}$ and $\phi_u^{SOP,LS}$ denote the network loss and SOP converter loss cost, respectively. $c_u^{SL}$ denotes the price of power loss in stage $u$, which is generally assigned as electricity price. $R_{ij}$ denotes the resistance of branch $ij$. $I_{ij,u}$ denotes the current magnitude of branch $ij$ in stage $u$. $P_{i,k,u}^{SOP,LS}$ denotes the active power loss of SOP converter at node $i$ in scheme $k$ in stage $u$.

## Constraints of FDN planning

The investment constraints of SOP are formulated as follows. The same terminal can only be used in one SOP planning scheme. When a scheme is determined in the previous stage, one of its evolutionary schemes should be selected in the next stage. The converter capacity of SOP is formulated as continuous variables for effective solutions. However, considering the modularisation requirements, the indeed installed SOP capacity is determined by rounding the corresponding variables in the solution.

$$\sum_{k\in\mathcal{L}_i}\alpha_{k,u} \le 1 \tag{10a}$$

$$\alpha_{k,u-1} \le \sum_{k'\in\mathcal{L}(k)}\alpha_{k',u} \tag{10b}$$

$$\sum_{k\in\mathcal{L}_i}S_{i,k,u-1}^{SOP} \le \sum_{k\in\mathcal{L}_i}S_{i,k,u}^{SOP} \tag{10c}$$

$$S_{k,u}^{SOP} = \sum_{i\in\Omega_k}S_{i,k,u}^{SOP} \tag{10d}$$

$$\pi\alpha_{k,u} \le S_{k,u}^{SOP} \le S_k^{SOP,max}\alpha_{k,u} \tag{10e}$$

where $S_{i,k,u}^{SOP}$ denotes the converter capacity at node $i$ in scheme $k$ in stage $u$. $S_k^{SOP,max}$ denotes the maximum capacity of SOP in scheme $k$. $\pi$ denotes a minimal positive value. The investment constraints of EVCS are formulated as follows.

$$\beta_{i,u-1} \le \beta_{i,u}, S_{i,u-1}^{EVCS} \le S_{i,u}^{EVCS} \tag{11a}$$

$$\pi\beta_{i,u} \le S_{i,u}^{EVCS} \le S_i^{EVCS,max}\beta_{i,u} \tag{11b}$$

$$P_{i,u}^{EV,rated} \le S_{i,u}^{EVCS} \tag{11c}$$

$$\sum_{i\in\Omega_e}P_{i,u}^{EV,rated} = P_u^{EV,pen} \tag{11d}$$

where $S_i^{EVCS,max}$ denotes the maximum capacity of EVCS at node $i$. $P_{i,u}^{EV,rated}$ denotes the rated EV demand at node $i$ in stage $u$. $P_u^{EV,pen}$

denotes the total EV demand in stage $u$. The investment constraints of PV are formulated as follows.

$$\delta_{i,u-1} \le \delta_{i,u}, S_{i,u-1}^{\mathrm{PV}} \le S_{i,u}^{\mathrm{PV}} \tag{12a}$$

$$\pi\delta_{i,u} \le S_{i,u}^{\mathrm{PV}} \le S_i^{\mathrm{PV,max}}\delta_{i,u} \tag{12b}$$

$$P_{i,u}^{\mathrm{PV,rated}} \le \mu_i^{\min} S_{i,u}^{\mathrm{PV}} \tag{12c}$$

$$\sum_{i\in\Omega_{\mathrm{g}}} P_{i,u}^{\mathrm{PV,rated}} = P_u^{\mathrm{PV,pen}} \tag{12d}$$

where $\delta_{i,u}$ is a binary variable, indicating whether PV is constructed at node $i$ in stage $u$. $S_i^{\mathrm{PV,max}}$ denotes the maximum capacity of PV at node $i$. $\mu_i^{\min}$ denotes the minimum power factor of PV at node $i$. $P_{i,u}^{\mathrm{PV,rated}}$ denotes the rated PV output at node $i$ in stage $u$. $P_u^{\mathrm{PV,pen}}$ denote the total PV output in stage $u$.

## Constraints of FDN operation

The power flow constraints of distribution network are formulated based on DistFlow branch model[40], which describes the power flow mechanism precisely and has been applied widely in distribution networks[41].

$$\sum_{ij\in\Omega_{\mathrm{b}}} \left( P_{ij,u} - R_{ij}I_{ij,u}^2 \right) + P_{j,u} = \sum_{jr\in\Omega_{\mathrm{b}}} P_{jr,u} \tag{13a}$$

$$\sum_{ij\in\Omega_{\mathrm{b}}} \left( Q_{ij,u} - X_{ij}I_{ij,u}^2 \right) + Q_{j,u} = \sum_{jr\in\Omega_{\mathrm{b}}} Q_{jr,u} \tag{13b}$$

$$V_{i,u}^2 - V_{j,u}^2 + \left( R_{ij}^2 + X_{ij}^2 \right)I_{ij,u}^2 - 2\left( R_{ij}P_{ij,u} + X_{ij}Q_{ij,u} \right) = 0 \tag{13c}$$

$$I_{ij,u}^2 V_{i,u}^2 - \left( P_{ij,u}^2 + Q_{ij,u}^2 \right) = 0 \tag{13d}$$

$$P_{i,u} = P_{i,u}^{\mathrm{S}} + P_{i,u}^{\mathrm{DG}} + \sum_{k\in\mathcal{L}} P_{i,k,u}^{\mathrm{SOP}} - P_{i,u}^{\mathrm{LD}} - P_{i,u}^{\mathrm{EV}} \tag{13e}$$

$$Q_{i,u} = Q_{i,u}^{\mathrm{S}} + Q_{i,u}^{\mathrm{DG}} + \sum_{k\in\mathcal{L}} Q_{i,k,u}^{\mathrm{SOP}} - Q_{i,u}^{\mathrm{LD}} \tag{13f}$$

where $\Omega_{\mathrm{b}}$ denotes the set of branches. $P_{ij,u}$ and $Q_{ij,u}$ denote the active and reactive power flow of branch $ij$ in stage $u$, respectively. $R_{ij}$ and $X_{ij}$ denote the resistance and reactance of branch $ij$, respectively. $V_{i,u}$ denotes the voltage magnitude of node $i$ in stage $u$. $P_{i,u}$ and $Q_{i,u}$ denote the total active and reactive power injection at node $i$ in stage $u$, respectively. $P_{i,u}^{\mathrm{S}}$, $Q_{i,u}^{\mathrm{S}}$, $P_{i,u}^{\mathrm{PV}}$, $Q_{i,u}^{\mathrm{PV}}$, $P_{i,u}^{\mathrm{LD}}$ and $Q_{i,u}^{\mathrm{LD}}$ denote the active and reactive power injection by substation, PV and load at node $i$ in stage $u$, respectively. $P_{i,k,u}^{\mathrm{SOP}}$ and $Q_{i,k,u}^{\mathrm{SOP}}$ denote the active and reactive power injection by SOP at node $i$ in scheme $k$ in stage $u$, respectively. $P_{i,u}^{\mathrm{EV}}$ denotes the active power injection by EV at node $i$ in stage $u$.

Note that the proposed FDN planning problem is a mixed-integer nonlinear programming (MINLP) model. A convex relaxation[42] is adopted to convert the MINLP model to a mixed-integer second-order conic programming (MISOCP) formulation, which can be efficiently computed by commercial solvers. The convex relaxation of power flow constraints is mathematically described as follows.

$$\sum_{ij\in\Omega_{\mathrm{b}}} \left( P_{ij,u} - R_{ij}l_{ij,u} \right) + P_{j,u} = \sum_{jr\in\Omega_{\mathrm{b}}} P_{jr,u} \tag{14a}$$

$$\sum_{ij\in\Omega_{\mathrm{b}}} \left( Q_{ij,u} - X_{ij}l_{ij,u} \right) + Q_{j,u} = \sum_{jr\in\Omega_{\mathrm{b}}} Q_{jr,u} \tag{14b}$$

$$v_{i,u} - v_{j,u} + \left( R_{ij}^2 + X_{ij}^2 \right)l_{ij,u} - 2\left( R_{ij}P_{ij,u} + X_{ij}Q_{ij,u} \right) = 0 \tag{14c}$$

$$\left\| \begin{array}{c} 2P_{ij,u,} \\ 2Q_{ij,u} \\ l_{ij,u} - v_{i,u} \end{array} \right\|_2 \le l_{ij,u} + v_{i,u} \tag{14d}$$

where $l_{ij,u}$ denotes the square of current magnitude of branch $ij$ in stage $u$. $v_{i,u}$ denotes the square of voltage magnitude of node $i$ in stage $u$. Namely, $l_{ij,u} = I_{ij,u}^2$ and $v_{i,u} = V_{i,u}^2$. To evaluate the accuracy of the convex relaxation for the proposed model, an index[43] is defined to quantify the relaxation deviation as follows.

$$J_u = \left\| l_{ij,u}v_{i,u} - P_{ij,u}^2 - Q_{ij,u}^2 \right\|_\infty \tag{15}$$

where $J_u$ denotes the index to evaluate the relaxation deviation, indicating whether the SOCP-relaxed optimal solution is accurate or not. If the gap is smaller than a pre-specified tolerance, the optimal solution is accepted as exact. Particularly, the demand of conventional loads and EVCSs, as well as the generation of PVs, are defined as random variables.

$$P_{i,u}^\vartheta = \xi_{i,u}^\vartheta \cdot P_{i,u}^{\vartheta,\mathrm{rated}}, \vartheta = \{\mathrm{LD,EV,PV}\} \tag{16}$$

where $\vartheta$ denotes the index of different devices. $\xi_{i,u}^\vartheta$ denotes the random profiles of device $\vartheta$ at node $i$ in stage $u$. The rated load power at node $i$ in stage $u$ is determined as $P_{i,u}^{\mathrm{LD,rated}} = P_{i,u-1}^{\mathrm{LD,rated}}(1+\rho_u^{\mathrm{LD}})^\Upsilon$, and $\rho_u^{\mathrm{LD}}$ denotes the annual increase rate of load in stage $u$. Assume the power factors of loads remain constant. The operational constraints of SOP are formulated as follows.

$$\sum_{i\in\Omega_k} \left( P_{i,k,u}^{\mathrm{SOP}} - P_{i,k,u}^{\mathrm{SOP,LOS}} \right) = 0 \tag{17a}$$

$$P_{i,k,u}^{\mathrm{SOP,LOS}} = \varpi\sqrt{\left( P_{i,k,u}^{\mathrm{SOP}} \right)^2 + \left( Q_{i,k,u}^{\mathrm{SOP}} \right)^2} \tag{17b}$$

$$S_{i,k,u}^{\mathrm{SOP}} \ge \sqrt{\left( P_{i,k,u}^{\mathrm{SOP}} \right)^2 + \left( Q_{i,k,u}^{\mathrm{SOP}} \right)^2} \tag{17c}$$

$$-S_{i,k,u}^{\mathrm{SOP}} \le P_{i,k,u}^{\mathrm{SOP}} \le S_{i,k,u}^{\mathrm{SOP}} \tag{17d}$$

$$-S_{i,k,u}^{\mathrm{SOP}} \le Q_{i,k,u}^{\mathrm{SOP}} \le S_{i,k,u}^{\mathrm{SOP}} \tag{17e}$$

where $\varpi$ denotes the loss factor of SOP converters. Chance constraints are formulated to represent the security risks of FDN with a predefined violation probability.

$$\mathbb{P}\left\{ V_{\min}^2 \le v_{i,u} \le V_{\max}^2 \right\} \ge 1-\gamma \tag{18a}$$

$$\mathbb{P}\left\{ l_{ij,u} \le I_{\max}^2 \right\} \ge 1-\gamma \tag{18b}$$

where $V_{\min}$ and $V_{\max}$ respectively denote the lower and upper bounds of nodal voltages, and $I_{\max}$ denotes the upper bound of line currents. $\gamma$ denotes the acceptable violation probability.

## Revised operational cost

After the planning strategy of FDN is determined, a large number of power flow calculations based on Monte Carlo method is executed to

compute the revised operational cost. The penalty cost is attributed to the load loss affected by potential security risks. Especially, the penalty cost for voltage violation is computed as the sum of the active power of the loads on the nodes where voltages exceed the safe range. The penalty cost for line overloading is computed as the sum of the active power of the loads located at the downstream nodes of the overload line. The calculation method is formulated as follows.

$$\phi_u^{\text{OP,all}} = 8760 \cdot \left( \phi_u^{\text{FDN,LS}} + \phi_u^{\text{VOLT,VL}} + \phi_u^{\text{CURT,VL}} \right) \tag{19a}$$

$$\phi_u^{\text{FDN,LS}} = c_u^{\text{SL}} \mathbb{E} \left[ \sum_{ij \in \Omega_b} R_{ij} l_{ij,u} + \sum_{k \in \mathcal{L}} \sum_{i \in \Omega_k} P_{i,k,u}^{\text{SOP,LS}} \right] \tag{19b}$$

$$\phi_u^{\text{VOLT,VL}} = c_u^{\text{VL}} \mathbb{E} \left[ \sum_{i \in \Omega_n} P_{i,u}^{\text{LD}} : v_{i,u} < V_{\min}^2 \parallel v_{i,u} > V_{\max}^2 \right] \tag{19c}$$

$$\phi_u^{\text{CURT,VL}} = c_u^{\text{VL}} \mathbb{E} \left[ \sum_{ij \in \Omega_b} \sum_{r \in \Omega_{n,ij}} P_{r,u}^{\text{LD}} : l_{ij,u} > I_{\max}^2 \right] \tag{19d}$$

where $\phi_u^{\text{FDN,LS}}$ denotes the cost of FDN loss. $\phi_u^{\text{VOLT,VL}}$ and $\phi_u^{\text{CURT,VL}}$ denote the costs of voltage violation and line overload, respectively. $\Omega_n$ denotes the set of nodes, and $\Omega_{n,ij}$ denotes the set of the nodes downstream of branch $ij$. $c_u^{\text{VL}}$ denotes the penalty price, which is generally assigned as the electricity price.

Additionally, the above penalty costs are not included in the objective of planning model, for the reason that the nodal voltages and line currents are restricted within a safe range in Eq. (18). However, the chance constrains allow violations within a permitted probability, so the costs of voltage violation and line overload are involved in the economic estimation of the planning scheme.

## Chance-constrained programming
The compact formulation of chance-constrained programming is expressed as follows.

$$\min_{x \in \chi} f(x) \tag{20a}$$

$$\boldsymbol{s}(\boldsymbol{x},\boldsymbol{w},\boldsymbol{\xi}) = 0 \tag{20b}$$

$$\boldsymbol{m}(\boldsymbol{x},\boldsymbol{w},\boldsymbol{\xi}) \leq 0 \tag{20c}$$

$$\mathbb{P}\{\boldsymbol{z}(\boldsymbol{x},\boldsymbol{w},\boldsymbol{\xi}) \in \mathcal{Z}\} \geq 1 - \boldsymbol{\gamma} \tag{20d}$$

where $\boldsymbol{x}$ denotes the vector of state variable. $\boldsymbol{w}$ denotes the vector of decision variables. $\boldsymbol{s}(\boldsymbol{x},\boldsymbol{w},\boldsymbol{\xi})$ and $\boldsymbol{m}(\boldsymbol{x},\boldsymbol{w},\boldsymbol{\xi})$ denote the equality and inequality constraints, respectively. $\boldsymbol{z}(\boldsymbol{x},\boldsymbol{w},\boldsymbol{\xi})$ is modelled as chance constraints, and $\mathcal{Z}$ denotes the feasible region determined by lower and upper limits $\boldsymbol{z}_{\min}$ and $\boldsymbol{z}_{\max}$.

With the adoption of Distflow constraints, the original problem defined in Eq. (20) is essentially a chance-constrained MISOCP model. However, there is no analytical expression for the chance constraints in a non-linear system. To obtain the violation probability of the distribution network, it is straightforward to use the sampling method. First, a sample set $\boldsymbol{\xi}_N = \left\{ \boldsymbol{\xi}^{(j)} \right\}_{j=1}^N$ is generated based on the modelling of uncertainties. Then, FDN states $\boldsymbol{x}_N = \left\{ \boldsymbol{x}^{(j)} \right\}_{j=1}^N$ are obtained when planning scheme $\boldsymbol{w}$ is adopted. Hence, the chance constraints can be

further expressed as follows.

$$\mathbb{P}\{z_i(\boldsymbol{x},\boldsymbol{w},\boldsymbol{\xi}) \in \mathcal{Z}_i\} = \mathbb{E}\left[\mathbb{I}\left(z_i(\boldsymbol{x}_N,\boldsymbol{w},\boldsymbol{\xi}_N)\right)\right] \tag{21a}$$

$$\mathbb{I}\left(z_i\left(\boldsymbol{x}^{(j)},\boldsymbol{w},\boldsymbol{\xi}^{(j)}\right)\right) = \begin{cases} 1 & z_i\left(\boldsymbol{x}^{(j)},\boldsymbol{w},\boldsymbol{\xi}^{(j)}\right) \in \mathcal{Z}_i \\ 0 & z_i\left(\boldsymbol{x}^{(j)},\boldsymbol{w},\boldsymbol{\xi}^{(j)}\right) \notin \mathcal{Z}_i \end{cases} \tag{21b}$$

where $\mathbb{I}(\cdot)$ denotes the signature function, and $\mathbb{E}[\cdot]$ denotes the expectation operator.

## General iterative format
The chance constrains in Eq. (20) are reformulated as follows using the expectation of random variables[28].

$$\begin{cases} \boldsymbol{z}(\boldsymbol{x},\boldsymbol{w},\mathbb{E}[\boldsymbol{\xi}]) \geq \boldsymbol{z}_{\min}^\kappa \\ \boldsymbol{z}(\boldsymbol{x},\boldsymbol{w},\mathbb{E}[\boldsymbol{\xi}]) \leq \boldsymbol{z}_{\max}^\kappa \end{cases} \tag{22a}$$

$$\begin{cases} \boldsymbol{z}_{\min}^\kappa = \boldsymbol{z}_{\min}^{\kappa-1} + \Delta\boldsymbol{z}_{\min}^{\kappa-1} \\ \boldsymbol{z}_{\max}^\kappa = \boldsymbol{z}_{\max}^{\kappa-1} - \Delta\boldsymbol{z}_{\max}^{\kappa-1} \end{cases} \tag{22b}$$

$$\begin{cases} \Delta\boldsymbol{z}_{\min}^\kappa = \mathbb{Q}(\boldsymbol{z},\mathbb{P}\{\boldsymbol{z} \in \mathcal{Z}\}) - \mathbb{Q}(\boldsymbol{z},\boldsymbol{\gamma}) \\ \Delta\boldsymbol{z}_{\max}^\kappa = \mathbb{Q}(\boldsymbol{z},1 - \boldsymbol{\gamma}) - \mathbb{Q}(\boldsymbol{z},\mathbb{P}\{\boldsymbol{z} \in \mathcal{Z}\}) \end{cases} \tag{22c}$$

where $\mathbb{Q}(\cdot)$ denotes the quantile function. $\Delta\boldsymbol{z}_{\min}^{\kappa-1}$ and $\Delta\boldsymbol{z}_{\max}^{\kappa-1}$ are both non-negative values, which are used to update the upper and lower bounds $\boldsymbol{z}_{\min}^\kappa$ and $\boldsymbol{z}_{\max}^\kappa$ of the inequality constraints, respectively. The initialisation conditions are $\boldsymbol{z}_{\min}^0 = \boldsymbol{z}_{\min}$, $\boldsymbol{z}_{\max}^0 = \boldsymbol{z}_{\max}$.

In this way, the chance-constrained optimisation model described in Eq. (20) is transformed into a deterministic model, which is solved with the updated bounds in each iteration until all the security constraints satisfy the predefined risk level. Constraints that occur outside of bounds are defined as valid constraints; otherwise, they are defined as invalid constraints. During the iteration process, only the corrections to the valid constraints need to be calculated and iteratively updated, whereas the invalid constraints remain unchanged. Therefore, using updated bounds to iteratively solve the planning scheme not only ensures a predetermined margin of safety, but also prevents the result from being overly conservative. Although this type of iterative algorithm does not have a convergence guarantee[44], it performs well in practical engineering applications.

## Modified iterative algorithm
In the general iteration format, there are two drawbacks to be improved. In the general iteration format, there are two limitations to be improved. First, the constraint bounds are updated from the previous bounds, as shown in Eq. (22b). However, the initial bounds are relatively relaxed, so that the bounds updated at the beginning of the iterations do not affect the solution, thus resulting in slow convergence. Hence, new bounds can be obtained in a straightforward manner by correcting the solution of the deterministic planning model, and Eq. (22b) can be rewritten as follows.

$$\begin{cases} \boldsymbol{z}_{\min}^\kappa = \min(\boldsymbol{z}_{\text{det}}^{\kappa-1} + \Delta\boldsymbol{z}_{\min}^{\kappa-1}, \boldsymbol{z}_{\max}^\kappa) \\ \boldsymbol{z}_{\max}^\kappa = \max(\boldsymbol{z}_{\text{det}}^{\kappa-1} - \Delta\boldsymbol{z}_{\max}^{\kappa-1}, \boldsymbol{z}_{\min}^\kappa) \end{cases} \tag{23}$$

where $\boldsymbol{z}_{\text{det}}^{\kappa-1}$ denotes the solution of deterministic planning model in the $(\kappa - 1)$-th iteration. The $\min(\cdot)$ and $\max(\cdot)$ operations are performed to avoid numerical conflicts between the upper and lower bounds. During the iterative process, only the corrections of the valid constraints need to be calculated and iteratively updated. The invalid constraint remains unchanged, thus satisfying $\boldsymbol{z}_{\min}^\kappa = \boldsymbol{z}_{\min}^{\kappa-1}$ and $\boldsymbol{z}_{\max}^\kappa = \boldsymbol{z}_{\max}^{\kappa-1}$.

The other limitation of the general iterative method is that at the end of the iterations, when the violation risk is adjacent to the

predetermined threshold, a smaller bound correction and slower convergence are resulted. Therefore, a penalised correction approach is proposed to ensure that the iterative process can be completed rapidly, and Eq. (22b) can be rewritten as follows.

$$\gamma_p^\kappa = \gamma \cdot e^{-\alpha^\kappa / \kappa_{max}} \tag{24a}$$

$$\begin{cases} \Delta z_{min}^\kappa = \mathbb{Q}(z, \mathbb{P}\{z \in \mathcal{Z}\}) - \mathbb{Q}\left(z, \gamma_p^\kappa\right) \\ \Delta z_{max}^\kappa = \mathbb{Q}\left(z, 1 - \gamma_p^\kappa\right) - \mathbb{Q}(z, \mathbb{P}\{z \in \mathcal{Z}\}) \end{cases} \tag{24b}$$

where $\kappa_{max}$ denotes the maximum number of iterations. $\alpha^\kappa$ denotes the cumulative number of times that the security constraints violate the risk assessment during iterations. Finally, the chance-constrained problem established in Eq. (20) is converted into a deterministic MISOCP model with an iterative format, which is formulated as follows. At each iteration, the model can be effectively solved by commercial solvers, such as Mosek or Gurobi.

$$\min_{x \in X} f(x)$$
$$\text{s.t.} (20b) - (20c), (20d), (23), (24) \tag{25}$$

### Reporting summary
Further information on research design is available in the Nature Portfolio Reporting Summary linked to this article.

## Data availability
The price data of grid assets over the entire planning period is available in the Supplementary Information file. The processed input data is sampled from the probabilistic distributions of sources and loads. The output data is generated by performing the multi-resource dynamic coordinated planning of flexible distribution network. Source data are provided with this paper.

## Code availability
The mathematical programming models are written by Python 3.7 and solved with the commercial solver Gurobi 10.0.1. Detailed descriptions of the sets, parameters, objective function, constraints, and variables are available in the Method section. Information about the code used in this research, including how to access it, are available on GitHub (https://github.com/fdn-planning/FDN_Model).

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

## Acknowledgements

This work is supported by National Natural Science Foundation of China "Clustering control of flexible distribution networks with large-scale DG integration" (No. U22B20114), "Trading mechanism and design of decentralized electric power market in distribution systems based on flexibility pricing" (No. 52277117), and the project of "Integrated operation and planning for smart electric distribution networks (OPEN)" from the UK and China. The researchers would like to acknowledge and thank the funders.

## Author contributions

R.W. and H.J. conceived the paper, wrote the code and drafted the manuscript. P.L. and H.Y. conceived the multi-resource dynamic and coordinated planning of flexible distribution network. J.Z. processed and analysed data. L.Z. collected and analysed data, and proofread the manuscript. Y.Z., J.W., L.B. and J.Y. edited and revised the manuscript. C.W. provided institutional and material support for the research.

## Competing interests

The authors declare no competing interests.
