## [Peer Review File · Nature Communications]

REVIEWER COMMENTS

Reviewer #1 (Remarks to the Author):

This paper proposes a planning algorithm for future distribution networks which contain soft-open-points to enable flexible power transfers. While the work is technically sound, the paper does not make a convincing case for its novelty. The primary innovation seems to be the modified iterative algorithm used to convert chance constraints to deterministic values which can be solved within the optimisation problem. While this is a valuable contribution, it is relatively marginal in the context of the overall field.

Further to this, there were some specific issues in how the paper is presented which currently represent barriers to its publication:

1. From whose perspective is the planning problem solved? The decision variables include the size and location of PV generation and EV charging infrastructure, which are generally outside the control of network companies.
2. Several of the figures, while very well presented, are difficult to interpret. In particular Fig 1 contains an illustration of a city which makes the differences in the network harder to identify and Fig 4 has illegibly small text in the legends.
3. No details are provided on how the electrical network has been modelled nor how the optimal power flow problem has been solved. This is highly unusual within the field, as this aspect of the problem is of high importance. Without this information it is not possible for someone to replicate the study and it is unclear how fast the solution will be, how accurate the model is, and whether a globally optimal solution can be obtained.

Overall I think the work in this paper probably is worthy of publication in a high impact journal, but the current manuscript does not present it well enough to be accepted in the current form.

Reviewer #2 (Remarks to the Author):

A coordinated dynamic plan for flexible distribution networks is described in the study. However, a few aspects are overlooked, which have an influence on the results:

- First, new PVs and EVCSs will be installed by prosumers in many places across the world (as far as I know). The DSO simply monitors these installations and their effects on the grid to determine whether they are feasible. Incentives for prosumers are thus the only tool for controlling the installation of new PVs and EVCSs in the grid. However, the DSO cannot have control over the location of new PVs and

EVCSs. Based on this, the paper's key contribution concerning the coordination of additional resources in FDN appears doubtful.

- Second, the operational penalty factor mentioned on page 5 is critical to the planning strategy. It is set at 100 CNY there. However, there is no reason for this. Furthermore, the mechanism for calculating this factor, which represents the penalty for overloads and overvoltages, is not described in the paper.

- Third, the operating cost due to overvoltage and overload is not stated in the proposed solution provided on page 11 (equation 7). This has the effect that the method is flawed and insufficient detail is supplied for the work to be reproduced.

- Fourth, alternatives to SOP are overlooked. In an FDN, for example, we can consider flexibility provided by storages, PVs, or demand responses. In many circumstances, the flexibility given by these resources will be more cost-effective than installing a SOP.

The paper also does not present what it promises at the end. For example,

- On page 4, it is stated that a cost-benefit analysis is carried out and compared to traditional planning techniques. However, on page 5, two basic cases that do not represent standard planning methodologies are addressed (one adding capacity to congested lines, which is no planning, and one doing nothing). Additional investigations are required to demonstrate the usefulness of the suggested strategy and algorithm for dynamic planning.

- On page 7, it is stated that Tables 2 and 3 present the outcomes of two planning methods. Tables 2 and 3 have no information in this respect.

- On page 9, the discussion section concludes that the FDN representation in honeycomb form may be a more sophisticated structure in the future. However, except in the discussion section, the honeycomb form of FDN is not mentioned elsewhere in the study. As a result, the work does not support the conclusion brought up in the discussion section.

Reviewer #3 (Remarks to the Author):

This paper presents a method for planning distribution networks that takes into account multiple resources and allows for long-term allocation strategies. It also addresses uncertainties in source-load relationships to mitigate security risks. However, there are several concerns that need to be addressed by the authors:

1. The paper does not explicitly mention its contributions. It would be helpful to have a dedicated paragraph for this.
2. The descriptions for cases I, II, and III of planning should be grouped together in the same paragraph.
3. It would be beneficial to investigate the simultaneous use of SOP and line reinforcement for cost reduction in planning by creating another case (case IV) and comparing the results.
4. The paper should provide a general description of the Nataf transformation and a reference.
5. In figure 8.a, the violation risk of the lower bound for voltages is shown, but it is unclear what exactly is being displayed. A description is needed to clarify this. The same applies to figures 8.b and 8.c.
6. The 1 percent loss coefficient for SOP seems highly optimistic and its impact on the system's total loss should be considered.
7. The paper should explain the logic behind the location designation for SOP and whether SOP capacity is involved in these schemes. Additionally, the step size for SOP capacity in the schemes should be addressed.
8. There are many symbols that are not introduced or introduced in later equations. For example, equations 5, γ , and Ω_{γ} are not introduced, and equations 6 and $\beta_{i,u}$ are not explained. A review and correction of these mistakes is necessary, and a nomenclature table may be useful.
9. The paper does not include the power flow model and related equations. At least a brief explanation of the model and power flow variables is required.
10. More explanation is needed for the method used to obtain equations 13, as it involves a large number of power flow calculations.
11. It may be better to move the solution procedure section to the beginning of the method part so that readers can have a general understanding of the solving procedure.
12. The paper does not mention the type of optimization problem or the solver used. If linearizations or relaxations have been implemented, they should be mentioned.

RESPONSES TO REVIEWER 1'S COMMENTS:

This paper proposes a planning algorithm for future distribution networks which contain soft-open-points to enable flexible power transfers. While the work is technically sound, the paper does not make a convincing case for its novelty. The primary innovation seems to be the modified iterative algorithm used to convert chance constraints to deterministic values which can be solved within the optimisation problem. While this is a valuable contribution, it is relatively marginal in the context of the overall field.

Overall I think the work in this paper probably is worthy of publication in a high impact journal, but the current manuscript does not present it well enough to be accepted in the current form. Further to this, there were some specific issues in how the paper is presented which currently represent barriers to its publication.

Authors' Response & Revision:

The authors wish to express our sincere gratitude for the insightful comments from the reviewer. We found all the comments and suggestions very helpful for further improving our paper. The authors have carefully addressed all the concerns raised by the reviewer.

The motivation behind this work is to explore and design a novel architecture of distribution networks based on soft open points (SOPs) with the integration of high penetration of distributed generators (DGs) and flexible loads. The paper highlights the flexible regulation and interconnection capabilities of SOPs in spatial dimension, which enables an interconnected and extensible architecture for distribution networks. Conventional distribution networks characterised by the radial and unidirectional structure are expected to evolve into flexible distribution networks (FDNs).

In the FDN, different feeders are flexibly interconnected by SOPs, then the power flow over the entire system can be controlled in closed-loop operation. With the dynamic planning techniques during a long-term period, the converter capacity and terminal number of SOP can expand with the coordinated planning of sources and loads. Thus, the investment cost can be reasonably assigned to each planning stage, which allows the consistency of FDN planning and avoids investment reset.

Therefore, the specific questions that we aim to address are, how to develop a successive FDN planning strategy over a long duration, and how to determine the siting and sizing of SOPs, electric vehicle charging stations (EVCSs) and photovoltaics (PVs) simultaneously, while considering source-load uncertainties.

The planning results can provide comprehensive guidance for future distribution network constructions. The contributions of this paper are summarised and further elaborated as follows.

1) A multi-resource dynamic planning method of FDNs is proposed, in which the configuration of SOPs, PVs and EVCSs is coordinated over a long-term planning period. The flexible reinforcement of FDN can be realised by multiple stages, and advantageous cost-benefits can be achieved compared with traditional planning approach.

2) In the FDN planning model, a probabilistic framework is established to address the strong source-load uncertainties. The security risks are formulated by chance constraints, and the stochastic nonlinear optimisation model is effectively solved based on the modified iterative algorithm. By adjusting the acceptable violation probability in chance constraints, a trade-off between investment efficiency and operational security can be obtained.

Comment 1.1

From whose perspective is the planning problem solved? The decision variables include the size and location of PV generation and EV charging infrastructure, which are generally outside the control of network companies.

Authors' Response & Revision:

Many thanks for the reviewer's suggestion. We agree that the planning and operation of PV generation and EV charging infrastructure may be outside the concern of power companies. Nevertheless, the coordinated planning method is proposed to maximise social benefits and achieve the overall economy efficiency for power companies, energy suppliers and users in distribution networks.

The paper aims to provide comprehensive planning guidance and suggestions for future distribution network constructions. The users or energy suppliers are not mandatory to follow the allocation result of sources and loads. But the user-built PVs and EV charging piles may bring operational violations to the distribution network, resulting in their own inability to plug into the grid. Therefore, it is beneficial to formulate planning guidance for individual users, especially for the energy suppliers that make profits by providing public generating and charging services. The PVs and EVCSs that constructed at the suggested positions will be better regulated and supported by SOPs. On the other hand, the flexible capabilities of

SOPs can be more fully utilised to ensure the safe hosting of renewable generation and charging load for power companies.

Furthermore, the proposed dynamic planning method enables the configuration of SOPs to adapt to the changes in source and load by stages. The expected capacities and available positions of household PVs and EV charging stations in each stage can be estimated, and be incorporated by the FDN coordinated planning model. With the simultaneous consideration of grid, source and load in a multi-stage planning framework, the planning guidance exhibits high compatibility for power companies, energy suppliers and users. As a result, the total investment for distribution network reinforcement can be reduced or delayed.

In summary, the coordinated planning of SOPs, EVCSs and PVs can not only proactively provide a comprehensive guidance for power companies, energy suppliers and users, but also accommodate the integrations of user-build PVs and EVCSs in a multi-stage planning framework. In this way, the construction of distribution networks can cover the interests of multiple parties and offer better power supply service in practical operations.

The detailed modifications are shown as follows.

“

The coordinated planning of SOPs, EVCSs and PVs to maximise the overall social benefits is investigated in this paper, which provides comprehensive planning guidance for power companies, energy suppliers and users in distribution networks. The users or energy suppliers are not mandatory to follow the allocation result of sources and loads. However, the user-built PVs and EV charging piles may cause operational violations, resulting in their own inability to plug into the grid [20]. Therefore, it is beneficial for users and energy suppliers to follow the planning guidance. With the simultaneous consideration of grid, source and load in a multi-stage planning model, the planning guidance exhibits high compatibility for all participants in the distribution network. In this way, the construction of distribution network can cover the interests of multiple parties and offer better power supply service in practical operations.

” (Section 1, page 2, highlighted in blue)

Comment 1.2

Several of the figures, while very well presented, are difficult to interpret. In particular Fig 1 contains an illustration of a city which makes the differences in the network harder to identify and Fig 4 has illegibly small text in the legends.

Authors' Response & Revision:

The authors would like to appreciate the respectable reviewer for the valuable suggestions. The illustration of distribution network in Fig.1 is highlighted and the flexible interconnections based on SOPs are emphasized. In addition, the layout of subgraphs in Fig.4 is optimized and the small texts are enlarged.

The detailed modifications are shown as follows.

“

Fig. 1. Illustration of conventional and flexible distribution networks with highly integrated DGs and EVs.

” (Section 1, page 2, highlighted in blue)

“

Fig. 4. Multi-resource dynamic coordinated planning scheme of FDN.

” (Section 4, page 5, highlighted in blue)

Comment 1.3

No details are provided on how the electrical network has been modelled nor how the optimal power flow problem has been solved. This is highly unusual within the field, as this aspect of the problem is of high importance. Without this information it is not possible for someone to replicate the study and it is unclear how fast the solution will be, how accurate the model is, and whether a globally optimal solution can be obtained.

Authors' Response & Revision:

We really appreciate the reviewer's comments. We have supplemented the power flow constraints of distribution network based on DistFlow branch model [40], as shown in (13). The DistFlow branch model describes the power flow mechanism precisely and has been applied widely in distribution networks [41].

Constraints (13.a) and (13.b) represent the active and reactive power balance of node i . Ohm's law over branch ij is expressed as (13.c). The current magnitude of each line can be determined by (13.d). Constraints (13.e) and (13.f) indicate the total active and reactive power injection of node i .

$$\sum_{ij \in \Omega_b} (P_{ij,u} - R_{ij} I_{ij,u}^2) + P_{j,u} = \sum_{jr \in \Omega_b} P_{jr,u} \quad (13.a)$$

$$\sum_{ij \in \Omega_b} (Q_{ij,u} - X_{ij} I_{ij,u}^2) + Q_{j,u} = \sum_{jr \in \Omega_b} Q_{jr,u} \quad (13.b)$$

$$V_{i,u}^2 - V_{j,u}^2 + (R_{ij}^2 + X_{ij}^2) I_{ij,u}^2 - 2(R_{ij} P_{ij,u} + X_{ij} Q_{ij,u}) = 0 \quad (13.c)$$

$$I_{ij,u}^2 V_{i,u}^2 - (P_{ij,u}^2 + Q_{ij,u}^2) = 0 \quad (13.d)$$

$$P_{i,u} = P_{i,u}^S + P_{i,u}^{PV} + \sum_{k \in \mathcal{L}} P_{i,k,u}^{SOP} - P_{i,u}^{LD} - P_{i,u}^{EV} \quad (13.e)$$

$$Q_{i,u} = Q_{i,u}^S + Q_{i,u}^{PV} + \sum_{k \in \mathcal{L}} Q_{i,k,u}^{SOP} - Q_{i,u}^{LD} \quad (13.f)$$

where $P_{ij,u}$ and $Q_{ij,u}$ denote the active and reactive power flow of branch ij in stage u , respectively. R_{ij} and X_{ij} denote the resistance and reactance of branch ij , respectively. $I_{ij,u}$ denotes the current magnitude of branch ij in stage u . $V_{i,u}$ denotes the voltage magnitude of node i in stage u .

Note that the proposed FDN planning problem is a mixed-integer nonlinear programming (MINLP) model. A convex relaxation [42] is adopted to convert the MINLP model to a mixed-integer second-order conic programming (MISOCP) formulation, which can be efficiently computed by commercial solvers. The convex relaxation of power flow constraints is mathematically described as follows.

$$\sum_{ij \in \Omega_b} (P_{ij,u} - R_{ij} l_{ij,u}) + P_{j,u} = \sum_{jr \in \Omega_b} P_{jr,u} \quad (14.a)$$

$$\sum_{ij \in \Omega_b} (Q_{ij,u} - X_{ij} l_{ij,u}) + Q_{j,u} = \sum_{jr \in \Omega_b} Q_{jr,u} \quad (14.b)$$

$$v_{i,u} - v_{j,u} + (R_{ij}^2 + X_{ij}^2) l_{ij,u} - 2(R_{ij} P_{ij,u} + X_{ij} Q_{ij,u}) = 0 \quad (14.c)$$

$$\left\| \begin{array}{c} 2P_{ij,u} \\ 2Q_{ij,u} \\ l_{ij,u} - v_{i,u} \end{array} \right\|_2 \leq l_{ij,u} + v_{i,u} \quad (14.d)$$

where $l_{ij,u}$ denotes the square of current magnitude of branch ij in stage u . $v_{i,u}$ denotes the square of voltage magnitude of node i in stage u . Namely, $l_{ij,u} = I_{ij,u}^2$ and $v_{i,u} = V_{i,u}^2$.

As for the solution speed of the MISOCP model, programs are executed is solved using Python 3.7 on a computer with an Intel Core i7-9700 3 GHz CPU and 64 GB RAM. Based on the investigations in case study, the model is solved within 49.93 minutes on average in each iteration, which satisfies the solution time requirements for long-term planning.

As for the solution precision, the convex relaxation in (14.d) is stated to be exact if the optimal solution obtained by solving the convexified problem is a feasible solution of the original non-convex problem. To evaluate the accuracy of the convex relaxation for the proposed model, an index [43] is defined to quantify the relaxation deviation as follows. In case studies, optimal solution is guaranteed with the maximum gap smaller than tolerance $1e-3$, which satisfies the solution accuracy for practical application.

$$J_u = \|l_{ij,u}v_{i,u} - P_{ij,u}^2 - Q_{ij,u}^2\|_{\infty} \quad (15)$$

where J_u denotes the index to evaluate the relaxation deviation, indicating whether the SOCP-relaxed optimal solution is accurate or not. If the gap is smaller than a pre-specified tolerance, the optimal solution is accepted as exact.

Further considering the uncertainties originating from distributed energy sources and loads, the security risks of FDN are formulated by chance constraints. As a result, the optimisation model is eventually extended to a chance-constrained MISOCP formulation. Using the proposed modified iterative algorithm, the chance constraints can be tractably solved.

The detailed modifications are shown as follows.

“

In each iteration, the deterministic optimisation model is solved within 49.93 minutes on average, and optimal solution is guaranteed with the maximum gap smaller than tolerance $1e-3$.

” (Section 6, page 9, highlighted in blue)

“

Constraints of FDN operation. The power flow constraints of distribution network are formulated based on DistFlow branch model [40], which describes the power flow mechanism precisely and has been applied widely in distribution networks [41].

$$\sum_{ij \in \Omega_b} (P_{ij,u} - R_{ij} I_{ij,u}^2) + P_{j,u} = \sum_{jr \in \Omega_b} P_{jr,u} \quad (13.a)$$

$$\sum_{ij \in \Omega_b} (Q_{ij,u} - X_{ij} I_{ij,u}^2) + Q_{j,u} = \sum_{jr \in \Omega_b} Q_{jr,u} \quad (13.b)$$

$$V_{i,u}^2 - V_{j,u}^2 + (R_{ij}^2 + X_{ij}^2) I_{ij,u}^2 - 2(R_{ij} P_{ij,u} + X_{ij} Q_{ij,u}) = 0 \quad (13.c)$$

$$I_{ij,u}^2 V_{i,u}^2 - (P_{ij,u}^2 + Q_{ij,u}^2) = 0 \quad (13.d)$$

$$P_{i,u} = P_{i,u}^S + P_{i,u}^{PV} + \sum_{k \in \mathcal{L}} P_{i,k,u}^{SOP} - P_{i,u}^{LD} - P_{i,u}^{EV} \quad (13.e)$$

$$Q_{i,u} = Q_{i,u}^S + Q_{i,u}^{PV} + \sum_{k \in \mathcal{L}} Q_{i,k,u}^{SOP} - Q_{i,u}^{LD} \quad (13.f)$$

where Ω_b denotes the set of branches. $P_{ij,u}$ and $Q_{ij,u}$ denote the active and reactive power flow of branch ij in stage u , respectively. R_{ij} and X_{ij} denote the resistance and reactance of branch ij , respectively. $V_{i,u}$ denotes the voltage magnitude of node i in stage u . $P_{i,u}$ and $Q_{i,u}$ denote the total active and reactive power injection at node i in stage u , respectively. $P_{i,u}^S$, $Q_{i,u}^S$, $P_{i,u}^{PV}$, $Q_{i,u}^{PV}$, $P_{i,u}^{LD}$ and $Q_{i,u}^{LD}$ denote the active and reactive power injection by substation, PV and load at node i in stage u , respectively. $P_{i,k,u}^{SOP}$ and $Q_{i,k,u}^{SOP}$ denote the active and reactive power injection by SOP at node i in scheme k in stage u , respectively. $P_{i,u}^{EV}$ denote the active power injection by EV at node i in stage u .

Note that the proposed FDN planning problem is a mixed-integer nonlinear programming (MINLP) model. A convex relaxation [42] is adopted to convert the MINLP model to a mixed-integer second-order conic programming (MISOCP) formulation, which can be efficiently computed by commercial solvers. The convex relaxation of power flow constraints is mathematically described as follows.

$$\sum_{ij \in \Omega_b} (P_{ij,u} - R_{ij} l_{ij,u}) + P_{j,u} = \sum_{jr \in \Omega_b} P_{jr,u} \quad (14.a)$$

$$\sum_{ij \in \Omega_b} (Q_{ij,u} - X_{ij} l_{ij,u}) + Q_{j,u} = \sum_{jr \in \Omega_b} Q_{jr,u} \quad (14.b)$$

$$v_{i,u} - v_{j,u} + (R_{ij}^2 + X_{ij}^2) l_{ij,u} - 2(R_{ij} P_{ij,u} + X_{ij} Q_{ij,u}) = 0 \quad (14.c)$$

$$\left\| \begin{array}{c} 2P_{ij,u} \\ 2Q_{ij,u} \\ l_{ij,u} - v_{i,u} \end{array} \right\|_2 \leq l_{ij,u} + v_{i,u} \quad (14.d)$$

where $l_{ij,u}$ denotes the square of current magnitude of branch ij in stage u . $v_{i,u}$ denotes the square of voltage magnitude of node i in stage u . Namely, $l_{ij,u} = I_{ij,u}^2$ and $v_{i,u} = V_{i,u}^2$.

To evaluate the accuracy of the convex relaxation for the proposed model, an index [43] is defined to quantify the relaxation deviation as follows.

$$J_u = \|l_{ij,u}v_{i,u} - P_{ij,u}^2 - Q_{ij,u}^2\|_{\infty} \quad (15)$$

where J_u denotes the index to evaluate the relaxation deviation, indicating whether the SOCP-relaxed optimal solution is accurate or not. If the gap is smaller than a pre-specified tolerance, the optimal solution is accepted as exact.

Particularly, the demand of conventional loads and EVCSs, as well as the generation of PVs, are defined as random variables.

$$P_{i,u}^{\kappa} = \xi_{i,u}^{\kappa} \cdot P_{i,u}^{\kappa,\text{rated}}, \kappa = \{\text{LD, EV, PV}\} \quad (16)$$

where κ denotes the index of different devices. $\xi_{i,u}^{\kappa}$ denotes the random profiles of device κ at node i in stage u . The rated load power at node i in stage u is determined as $P_{i,u}^{\text{LD,rated}} = P_{i,u-1}^{\text{LD,rated}}(1 + \rho_u^{\text{LD}})^Y$, and ρ_u^{LD} denotes the annual increase rate of load in stage u . Assume the power factors of loads remain constant.

” (Section 9, page 14, highlighted in blue)

“

[40] Baran, M.E. & Wu, F.F. Optimal capacitor placement on radial distribution systems. IEEE Trans. Power Deliv. 4(1), 725–734 (1989)

[41] Daniel, K.M., Florian, D., Henrik, S., Steven, H.L., Sambuddha, C., Ross, B. & Javad, L. A survey of distributed optimization and control algorithms for electric power systems. IEEE Trans. Smart Grid 8(6), 2941–2962 (2017)

[42] Lavaei, J. & Low, S.H. Zero duality gap in optimal power flow problem. IEEE Trans. Power Syst. 27(1), 92–107 (2012)

[43] Wei, W., Wang, J., Li, N. & Mei, S. Optimal Power flow of radial networks and its variations: a sequential convex optimization approach. IEEE Trans. Smart Grid 8(6), 2974-2987 (2017)

” (Reference, page 20, highlighted in blue)

RESPONSES TO REVIEWER 2'S COMMENTS:

A coordinated dynamic plan for flexible distribution networks is described in the study. However, a few aspects are overlooked, which have an influence on the results.

Authors' Response & Revision:

The authors would like to appreciate the respectable reviewer for the valuable suggestions. We found all the comments and suggestions are very helpful for further improving our paper. The authors wish to express our sincere gratitude for the insightful comments from the reviewer. The authors have addressed all the reviewer's comments carefully.

Comment 2.1

First, new PVs and EVCSs will be installed by prosumers in many places across the world (as far as I know). The DSO simply monitors these installations and their effects on the grid to determine whether they are feasible. Incentives for prosumers are thus the only tool for controlling the installation of new PVs and EVCSs in the grid. However, the DSO cannot have control over the location of new PVs and EVCSs. Based on this, the paper's key contribution concerning the coordination of additional resources in FDN appears doubtful.

Authors' Response & Revision:

Many thanks for the reviewer's suggestion. The coordinated planning method is proposed to maximise social benefits and achieve the overall economy efficiency for power companies, energy suppliers and users in distribution networks.

The paper aims to provide comprehensive planning guidance and suggestions for future distribution network constructions. The users or energy suppliers are not mandatory to follow the allocation result of sources and loads. But the user-built PVs and EV charging piles may bring operational violations to the distribution network, resulting in their own inability to plug into the grid. Therefore, it is beneficial to formulate planning guidance for individual users, especially for the energy suppliers that make profits by providing public generating and charging services. The PVs and EVCSs that constructed at the suggested positions will be better regulated and supported by SOPs. On the other hand, the flexible capabilities of

SOPs can be more fully utilised to ensure the safe hosting of renewable generation and charging load for power companies.

Furthermore, the proposed dynamic planning method enables the configuration of SOPs to adapt to the changes in source and load by stages. The expected capacities and available positions of household PVs and EV charging stations in each stage can be estimated, and be incorporated by the FDN coordinated planning model. With the simultaneous consideration of grid, source and load in a multi-stage planning framework, the planning guidance exhibits high compatibility for power companies, energy suppliers and users. As a result, the total investment for distribution network reinforcement can be reduced or delayed.

In summary, the coordinated planning of SOPs, EVCSs and PVs can not only proactively provide a comprehensive guidance for power companies, energy suppliers and users, but also accommodate the integrations of user-build PVs and EVCSs in a multi-stage planning framework. In this way, the construction of distribution networks can cover the interests of multiple parties and offer better power supply service in practical operations.

The detailed modifications are shown as follows.

“

The coordinated planning of SOPs, EVCSs and PVs to maximise the overall social benefits is investigated in this paper, which provides comprehensive planning guidance for power companies, energy suppliers and users in distribution networks. The users or energy suppliers are not mandatory to follow the allocation result of sources and loads. However, the user-built PVs and EV charging piles may cause operational violations, resulting in their own inability to plug into the grid [20]. Therefore, it is beneficial for users and energy suppliers to follow the planning guidance. With the simultaneous consideration of grid, source and load in a multi-stage planning model, the planning guidance exhibits high compatibility for all participants in the distribution network. In this way, the construction of distribution network can cover the interests of multiple parties and offer better power supply service in practical operations.

” (Section 1, page 2, highlighted in blue)

Comment 2.2

Second, the operational penalty factor mentioned on page 5 is critical to the planning strategy. It is set at 100 CNY there. However, there is no reason for this. Furthermore, the mechanism for calculating this factor, which represents the penalty for overloads and overvoltages, is not described in the paper.

Authors' Response & Revision:

The authors would like to appreciate the respectable reviewer for reasonable concerns. The rationality of calculating penalty cost has been improved and elaborated as follows.

The penalty cost is attributed to the load loss affected by potential security risks. Especially, the penalty cost for voltage violation is computed as the sum of the active power of the loads on the nodes where voltages exceed the safe range. The penalty cost for line overloading is computed as the sum of the active power of the loads located at the downstream nodes of the overload line. The calculation method is formulated as follows. Particularly, the penalty price for unit load power is generally assigned as the electricity price in this paper.

$$\phi_u^{\text{OP,all}} = 8760 \cdot (\phi_u^{\text{FDN,LS}} + \phi_u^{\text{VOLT,VL}} + \phi_u^{\text{CURT,VL}}) \quad (19.a)$$

$$\phi_u^{\text{FDN,LS}} = c_u^{\text{SL}} \mathbb{E} [\sum_{ij \in \Omega_b} R_{ij} l_{ij,u} + \sum_{k \in \mathcal{L}} \sum_{i \in \Omega_k} P_{i,k,u}^{\text{SOP,LS}}] \quad (19.b)$$

$$\phi_u^{\text{VOLT,VL}} = c_u^{\text{VL}} \mathbb{E} [\sum_{i \in \Omega_n} P_{i,u}^{\text{LD}} : v_{i,u} < V_{\min}^2 || v_{i,u} > V_{\max}^2] \quad (19.c)$$

$$\phi_u^{\text{CURT,VL}} = c_u^{\text{VL}} \mathbb{E} [\sum_{ij \in \Omega_b} \sum_{r \in \Omega_{n,ij}} P_{r,u}^{\text{LD}} : l_{ij,u} > I_{\max}^2] \quad (19.d)$$

where $\phi_u^{\text{FDN,LS}}$ denotes the cost of FDN loss. $\phi_u^{\text{VOLT,VL}}$ and $\phi_u^{\text{CURT,VL}}$ denote the costs of voltage violation and line overload, respectively. Ω_n denotes the set of nodes, and $\Omega_{n,ij}$ denotes the set of the nodes downstream of branch ij . c_u^{VL} denotes the penalty price.

Additionally, the above penalty costs are not included in the objective of planning model, for the reason that the nodal voltages and line currents are restricted within a safe range in Eq. (18). However, the chance constrains allow violations within a permitted probability, so the costs of voltage violation and line overload are involved in the economic estimation of the planning scheme.

$$\mathbb{P}\{V_{\min}^2 \leq v_{i,u} \leq V_{\max}^2\} \geq 1 - \gamma \quad (18.a)$$

$$\mathbb{P}\{l_{ij,u} \leq I_{\max}^2\} \geq 1 - \gamma \quad (18.b)$$

where V_{\min} and V_{\max} respectively denote the lower and upper bounds of nodal voltages, and I_{\max} denotes the upper bound of line currents. γ denotes the acceptable violation probability.

The detailed modifications are shown as follows.

“

Revised operational cost. After the planning strategy of FDN is determined, a large number of power flow calculations based on Monte Carlo method is executed to compute the revised operational cost.

The penalty cost is attributed to the load loss affected by potential security risks. Especially, the penalty cost for voltage violation is computed as the sum of the active power of the loads on the nodes where voltages exceed the safe range. The penalty cost for line overloading is computed as the sum of the active power of the loads located at the downstream nodes of the overload line. The calculation method is formulated as follows.

$$\phi_u^{\text{OP,all}} = 8760 \cdot (\phi_u^{\text{FDN,LS}} + \phi_u^{\text{VOLT,VL}} + \phi_u^{\text{CURT,VL}}) \quad (19.a)$$

$$\phi_u^{\text{FDN,LS}} = c_u^{\text{SL}} \mathbb{E}[\sum_{ij \in \Omega_b} R_{ij} l_{ij,u} + \sum_{k \in \mathcal{L}} \sum_{i \in \Omega_k} P_{i,k,u}^{\text{SOP,LS}}] \quad (19.b)$$

$$\phi_u^{\text{VOLT,VL}} = c_u^{\text{VL}} \mathbb{E}[\sum_{i \in \Omega_n} P_{i,u}^{\text{LD}} : v_{i,u} < V_{\min}^2 \mid v_{i,u} > V_{\max}^2] \quad (19.c)$$

$$\phi_u^{\text{CURT,VL}} = c_u^{\text{VL}} \mathbb{E}[\sum_{ij \in \Omega_b} \sum_{r \in \Omega_{n,ij}} P_{r,u}^{\text{LD}} : l_{ij,u} > I_{\max}^2] \quad (19.d)$$

where $\phi_u^{\text{FDN,LS}}$ denotes the cost of FDN loss. $\phi_u^{\text{VOLT,VL}}$ and $\phi_u^{\text{CURT,VL}}$ denote the costs of voltage violation and line overload, respectively. Ω_n denotes the set of nodes, and $\Omega_{n,ij}$ denotes the set of the nodes downstream of branch ij . c_u^{VL} denotes the penalty price, which is generally assigned as the electricity price.

Additionally, the above penalty costs are not included in the objective of planning model, for the reason that the nodal voltages and line currents are restricted within a safe range in Eq. (18). However, the chance

constrains allow violations within a permitted probability, so the costs of voltage violation and line overload are involved in the economic estimation of the planning scheme.

” (Section 9, page 15, highlighted in blue)

Comment 2.3

Third, the operating cost due to overvoltage and overload is not stated in the proposed solution provided on page 11 (equation 7). This has the effect that the method is flawed and insufficient detail is supplied for the work to be reproduced.

Authors' Response & Revision:

The authors would like to thank the reviewer for the valuable comments for further improving the paper's quality. As elaborated in Response 2.2, the penalty costs are not included in the objective of planning model, for the reason that the nodal voltages and line currents are restricted within a safe range in Eq. (18). However, the chance constrains allow violations within a permitted probability, so the costs of voltage violation and line overload are involved in the operational cost when the planning scheme is applied in practice.

$$\mathbb{P}\{V_{\min}^2 \leq v_{i,u} \leq V_{\max}^2\} \geq 1 - \gamma \quad (18.a)$$

$$\mathbb{P}\{l_{ij,u} \leq I_{\max}^2\} \geq 1 - \gamma \quad (18.b)$$

where V_{\min} and V_{\max} respectively denote the lower and upper bounds of nodal voltages, and I_{\max} denotes the upper bound of line currents. γ denotes the acceptable violation probability.

The detailed modifications are shown as follows.

“

Additionally, the above penalty costs are not included in the objective of planning model, for the reason that the nodal voltages and line currents are restricted within a safe range in Eq. (18). However, the chance constrains allow violations within a permitted probability, so the costs of voltage violation and line overload are involved in the economic estimation of the planning scheme.

” (Section 9, page 15, highlighted in blue)

Comment 2.4

Fourth, alternatives to SOP are overlooked. In an FDN, for example, we can consider flexibility provided by storages, PVs, or demand responses. In many circumstances, the flexibility given by these resources will be more cost-effective than installing a SOP.

Authors' Response & Revision:

Many thanks for the reviewer's comment. We agree that other controllable resources, such as storages, distributed generators and demand responses, can also provide flexibility for distribution networks.

This paper highlights the advantage of SOP in spatial interconnections, which enables the radial conventional distribution network to gradually evolve into a flexibly interconnected and extensible architecture. The converter capacity of SOPs can be installed in one site, but shared by different feeders, which is beneficial to save the cost for site occupancy and land construction. Based on multi-terminal SOPs, power flow over the entire system can be dispatched in closed-loop operation, and the resources located in different areas can be coordinated more efficiently.

These advantages cannot be realised in traditional distribution networks, and other controllable resources have little impact and contribution on the structure evolution of power distribution grid. Therefore, SOP takes part as the key infrastructure for the structure evolution of distribution networks, which is the priority to be considered in the FDN planning.

In case study, energy storage systems (ESSs) planning is designed as Case III for cost-benefit analysis. In Case III, ESSs are planned with the coordination of EVCSs and PVs to ensure the safe operation of the distribution network. The detailed planning result and cost are given as follows. The allowed minimum and maximum state of charge (SOC) of ESSs are set to 10% and 90%. The storage battery price is set to 1000.0, 800.0, 500.0 and 300.0 CNY/kWh in Stage I-IV, respectively. Other prices, especially the prices for converter investment and land construction, are consistent with planning SOPs in Case I.

Table 1 Planning results of Case III

Stage	ESS allocation: position (capacity /MVA)		EVCS allocation: position (capacity /MVA)			PV allocation: position (capacity /MVA)		
	Converter	Battery						
I	-	-	49(1.65)			10(2.87) 24(0.45) 39(0.79) 40(0.04) 55(0.79) 64(0.31)		
II	7(0.16) 83(0.50)	83(0.35)	28(2.00) 49(2.00) 68(1.96)			10(3.00) 24(1.63) 29(1.92) 39(1.81) 40(0.04) 55(2.54) 60(0.15) 63(0.70) 64(0.55) 76(2.23)		
III	7(0.31) 72(0.35) 83(0.67)	83(0.54)	28(2.00) 49(2.00) 68(2.00) 72(0.91) 82(1.96)			10(3.00) 11(3.00) 24(2.78) 29(3.00) 39(2.81) 40(0.04) 55(3.00) 60(1.52) 63(0.74) 64(0.58) 76(2.88)		
IV	7(0.40) 39(1.23) 64(3.04) 72(0.79) 83(1.57)	39(1.42) 64(3.60) 72(0.64) 83(1.49)	28(2.00) 49(2.00) 68(2.00) 72(2.00) 82(1.96) 83(1.53)			10(3.00) 11(3.00) 24(3.00) 29(3.00) 39(3.00) 40(3.00) 55(3.00) 60(2.72) 63(3.00) 64(3.00) 76(3.00)		

Table 2 Planning cost of Case III

Stage	Investment cost (10 ⁶ CNY)						Operational cost (10 ⁶ CNY)			Sum (10 ⁶ CNY)
	ESS			EVCS		PV	System loss	Line overload	Voltage violation	
	Land exploitation	Converter	Battery	Land exploitation	Converter	Converter				
I	0.00	0.00	0.00	3.00	1.32	4.20	2.36	0.11	0.66	11.64
II	4.35	0.25	0.17	4.35	1.61	2.73	1.36	2.20	0.72	17.72
III	1.54	0.10	0.04	3.08	0.45	1.66	0.82	2.65	0.58	10.93
IV	2.39	0.27	0.47	1.20	0.13	0.45	0.56	1.42	0.16	7.06
Total		9.59		15.13		9.04	5.11	6.38	2.12	47.36

The ESS investment is used to build the site and purchase converters and batteries. To simplify calculations, assume that the battery of ESS can be charged or discharged by its maximum available capacity, and the sequential energy constraints are not considered, which will result in a smaller capacity investment cost of battery. However, confined to the radial structure of distribution network, ESSs need to be installed independently at each overrunning feeder, thus causing higher site construction cost. The total cost of planning ESSs is larger than that of planning SOPs in Case I, indicating that the flexible interconnected structure based on SOPs is economically promising for it offers resource sharing among feeders.

The descriptions for cases are given as follows, and the costs of all cases are illustrated in Fig. 5.

Case I: The multi-resource dynamic coordinated planning of FDN is performed.

Case II: The coordinated FDN planning is performed without stage division.

Case III: The energy storage systems (ESSs) planning is performed in the distribution network.

Case IV: The traditional planning method is performed, where the larger-capacity lines and transformers are invested for overloaded feeders.

Case V: The distribution network is not reinforced with the increase of sources and loads.

Fig. 5. Costs of different planning cases. Under a moderate investment cost for establishing a flexible interconnected structure based on SOPs, the operational penalty cost for voltage violations and line overloads can be reduced. Compared with Case II, the multi-stage planning framework can delay investment. Compared with Case III, the resource sharing is promising based on the flexible interconnected structure. Compared with Case IV, the proposed planning method exhibits better flexibility for expansion. Compared with Case V, the proposed planning method effectively address the security issue caused by the integration of PVs and EVCSs.

As for the coordinated planning, the siting and sizing of PV generation and EV charging demand are both optimised. Energy storages and demand responses can be further considered in subsequent research. Furthermore, SOP is compatible with other resources, for instance, the energy storage can be integrated into the DC link of SOP, which can be studied in future works.

The detailed modifications are shown as follows.

“

In Case III, ESSs are planned to ensure the safe operation of the distribution network. The ESS investment is used to build the site and purchase converters and batteries. The allowed minimum and maximum state of charge (SOC) of ESSs are set to 10% and 90%. Assume that the battery of ESS can be charged or discharged by its maximum available capacity, and the sequential energy constraints are not considered, which will result in a smaller capacity investment cost of battery. However, confined to the radial structure of distribution network, ESSs need to be installed independently at each overrunning

feeder, thus causing higher site construction cost. The total cost of planning ESSs is larger than that of planning SOPs in Case I, indicating that the flexible interconnected structure based on SOPs is economically promising for it offers resource sharing among feeders.

” (Section 5, page 6, highlighted in blue)

“

Fig. 5. Costs of different planning cases. Under a moderate investment cost for establishing a flexible interconnected structure based on SOPs, the operational penalty cost for voltage violations and line overloads can be reduced. Compared with Case II, the multi-stage planning framework can delay investment. Compared with Case III, the resource sharing is promising based on the flexible interconnected structure. Compared with Case IV, the proposed planning method exhibits better flexibility for expansion. Compared with Case V, the proposed planning method effectively address the security issue caused by the integration of PVs and EVCSs.

” (Section 5, page 7, highlighted in blue)

“

SOP takes part as the key infrastructure for the structure evolution of distribution networks, which is the priority to be considered in the FDN planning. Other controllable resources, such as energy storages and demand responses, can be further considered in subsequent research. In addition, the common DC bus of SOP is a compatible interface that provides access to DC loads, DC sources, and energy storage systems. In summary, FDN is both eco-friendly and economical to host emerging elements with diverse characteristics.

” (Section 8, page 10, highlighted in blue)

Comment 2.5

The paper also does not present what it promises at the end. For example, on page 4, it is stated that a cost-benefit analysis is carried out and compared to traditional planning techniques. However, on page 5, two basic cases that do not represent standard planning methodologies are addressed (one adding capacity to congested lines, which is no planning, and one doing nothing). Additional investigations are required to demonstrate the usefulness of the suggested strategy and algorithm for dynamic planning.

Authors' Response & Revision:

Many thanks for the careful review from the reviewer. The conclusion of the paper and the elaboration of case settings are further provided. In this paper, a multi-resource dynamic planning method of FDNs is proposed, and a probabilistic framework is established to address source-load uncertainties. Based on the investigations in case study, the configuration of SOPs, PVs and EVCSs is coordinated over a long-term planning period. The operational violations are effectively mitigated and advantageous cost-benefits are achieved compared with traditional planning approach.

To better analyse the cost benefit of the proposed planning method, two additional cases are supplemented. The detailed descriptions are given as follows.

Case I: The multi-resource dynamic coordinated planning of FDN is performed.

Case II: The coordinated FDN planning is performed without stage division.

Case III: The energy storage systems (ESSs) planning is performed in the distribution network.

Case IV: The traditional planning method is performed, where the larger-capacity lines and transformers are invested for overloaded feeders.

Case V: The distribution network is not reinforced with the increase of sources and loads.

In Case I, with the coordinated allocation of PVs and EVCSs, an interconnected architecture based on multi-terminal SOPs is formulated to provide spatial flexibility.

In Case II, the FDN planning is conducted within one stage, where the network evolution and the gradual growth in equipment capacity are not considered. The costs are calculated at current prices, and the investments are paid at the initial to meet the needs over the entire planning period. The total cost is the highest at 90.48×10^6 CNY.

In Case III, ESSs are planned to ensure the safe operation of the distribution network. The ESS investment is used to build the site and purchase converters and batteries. The allowed minimum and maximum state of charge (SOC) of ESSs are set to 10% and 90%. Assume that the battery of ESS can be charged or discharged by its maximum available capacity, and the sequential energy constraints are not considered, which will result in a smaller capacity investment cost of battery. However, confined to the radial structure of distribution network, ESSs need to be installed independently at each overrunning feeder, thus causing higher site construction cost. The detailed planning result are given in Response 2.4. The total cost of planning ESSs is larger than that of planning SOPs in Case I, indicating that the flexible interconnected structure based on SOPs is economically promising for it offers resource sharing among feeders.

In Case IV, the feeders that violate security criterion in each stage are identified to be reinforced. In this case, Feeders A and I in Stage II, Feeder K in Stage III, and Feeders E and H in Stage IV are reinforced. Additionally, the transformer capacity of the reinforced feeder needs to be expanded as well. The operational risk of the distribution network is reduced by planning larger-capacity lines and transformers. However, inadequate flexibility in capacity allocation leads to higher expansion costs. The total cost reaches 43.63×10^6 CNY. The mathematical formulations are given in Eq. (2.1).

$$\begin{aligned}
 \phi_u^{\text{Feeder,BR}} &= c_u^{\text{RF}} (\sum_{i \in \Omega_f} L_i \sigma_{i,u} - \sum_{i \in \Omega_f} L_i \sigma_{i,u-1}) \\
 \phi_u^{\text{Feeder,TR}} &= c_u^{\text{TR}} (\sum_{i \in \Omega_f} S_{i,u}^{\text{Feeder}} - \sum_{i \in \Omega_f} S_{i,u-1}^{\text{Feeder}}) \\
 \sigma_{i,u-1} &\leq \sigma_{i,u}, \quad S_{i,u-1}^{\text{Feeder}} \leq S_{i,u}^{\text{Feeder}} \\
 \sum_{j \in \Omega_{f,i}} P_{j,u}^{\text{LD}} &\leq S_{i,u}^{\text{Feeder}}, \quad \forall i \in \Omega_f
 \end{aligned} \tag{2.1}$$

where $\phi_u^{\text{Feeder,BR}}$ and $\phi_u^{\text{Feeder,TR}}$ denote the line and transformer reinforcement costs in stage u , respectively. c_u^{RF} and c_u^{TR} denote the prices of line and transformer reinforcement in stage u , respectively. Ω_f denotes the set of feeders, and $\Omega_{f,i}$ denotes the set of nodes in feeder i . L_i denotes the length of feeder i . $S_{i,u}^{\text{Feeder}}$ denotes the transmission capacity of the feeder i in stage u . $\sigma_{i,u}$ is a binary variable, indicating whether feeder i is reinforced in stage u .

In Case V, not implementing planning measures saves the investment cost, but voltage violations and line overloads severely jeopardise the safe operation of the distribution network, thus resulting in a higher penalty cost.

In summary, a dynamic coordinated FDN planning method is adopted in Case I, which alleviates operational risks while maintaining a lower investment cost. The total cost reduces by 55.02%, 14.06%, 6.72%, and 23.08% as compared with those of Case II-V, respectively. The results indicate that the flexible upgrading based on SOPs in a multi-stage framework offers better economy efficiency for managing security risks in distribution networks. The costs of different planning schemes are illustrated in Fig. 5.

Fig. 5. Costs of different planning cases. Under a moderate investment cost for establishing a flexible interconnected structure based on SOPs, the operational penalty cost for voltage violations and line overloads can be reduced. Compared with Case II, the multi-stage planning framework can delay investment. Compared with Case III, the resource sharing is promising based on the flexible interconnected structure. Compared with Case IV, the proposed planning method exhibits better flexibility for expansion. Compared with Case V, the proposed planning method effectively address the security issue caused by the integration of PVs and EVCSs.

The detailed modifications are shown as follows.

“

Planning strategy formulation. With the consideration of source-load uncertainties, five cases are designed for FDN planning, and their planning results and cost-benefits are further analysed.

Case I: The multi-resource dynamic coordinated planning of FDN is performed.

Case II: The coordinated FDN planning is performed without stage division.

Case III: The energy storage systems (ESSs) planning is performed in the distribution network.

Case IV: The traditional planning method is performed, where the larger-capacity lines and transformers are invested for overloaded feeders.

Case V: The distribution network is not reinforced with the increase of sources and loads.

” (Section 4, page 5, highlighted in blue)

“

Cost-benefit analysis. The economic efficiency of above cases in alleviating security risks are further investigated. The costs of different planning schemes are illustrated in Fig. 5.

In Case II, the FDN planning is conducted within one stage, where the network evolution and the gradual growth in equipment capacity are not considered. The costs are calculated at current prices, and the investments are paid at the initial to meet the needs over the entire planning period. The total cost is the highest at 90.48×10^6 CNY.

In Case III, ESSs are planned to ensure the safe operation of the distribution network. The ESS investment is used to build the site and purchase converters and batteries. The allowed minimum and maximum state of charge (SOC) of ESSs are set to 10% and 90%. Assume that the battery of ESS can be charged or discharged by its maximum available capacity, and the sequential energy constraints are not considered, which will result in a smaller capacity investment cost of battery. However, confined to the radial structure of distribution network, ESSs need to be installed independently at each overrunning feeder, thus causing higher site construction cost. The total cost of planning ESSs is larger than that of planning SOPs in Case I, indicating that the flexible interconnected structure based on SOPs is economically promising for it offers resource sharing among feeders.

In Case IV, the feeders that violate security criterion in each stage are identified to be reinforced. In this case, Feeders A and I in Stage II, Feeder K in Stage III, and Feeders E and H in Stage IV are reinforced. The maximum capacity of the expanded line is 1.5 times that of the original line, and the cost of expanding the line per unit length is thrice that of line construction [32]. Additionally, the transformer capacity of the reinforced feeder needs to be expanded as well. The operational risk of the distribution network is reduced by planning larger-capacity lines and transformers. However, inadequate flexibility in capacity allocation leads to higher expansion costs. The total cost reaches 43.63×10^6 CNY.

” (Section 5, page 6, highlighted in blue)

“

Fig. 5. Cost-benefit of different planning cases. Under a moderate investment cost for establishing a flexible interconnected structure based on SOPs, the operational penalty cost for voltage violations and line overloads can be reduced. Compared with Case II, the multi-stage planning framework can delay investment. Compared with Case III, the resource sharing is promising based on the flexible interconnected structure. Compared with Case IV, the proposed planning method exhibits better flexibility for expansion. Compared with Case V, the proposed planning method effectively address the security issue caused by the integration of PVs and EVCSs.

” (Section 5, page 7, highlighted in blue)

Comment 2.6

On page 7, it is stated that Tables 2 and 3 present the outcomes of two planning methods. Tables 2 and 3 have no information in this respect.

Authors' Response & Revision:

Many thanks for the reviewer's kind suggestion. The statements related to Tables 2 and 3 in page 7 are improved to be clearer and more intelligent. Tables 2 and 3 are respectively illustrated to show the planning scheme and cost of Case I, which are obtained by solving the proposed dynamic planning model based on the modified iterative algorithm. In Section 6 (on page 7), it is the performance of different algorithms used to solve the planning model that is compared. Additionally, the outcomes of different planning methods can be found in Sections 4 and 5.

The detailed modifications are shown as follows.

“

To solve the stochastic optimisation for FDN planning, the modified iterative algorithm is proposed in the paper. To analyse the performance of the solution algorithm, a comparison between the general and the modified iterative algorithm is conducted. The solution results obtained by the two algorithms are almost the same, but the general algorithm necessitates 21 iterations to converge, which requires 57.46 hours. However, using the modified iterative algorithm, only 6 iterations are necessitated to attain convergence, which requires 9.40 hours.

” (Section 6, page 9, highlighted in blue)

Comment 2.7

On page 9, the discussion section concludes that the FDN representation in honeycomb form may be a more sophisticated structure in the future. However, except in the discussion section, the honeycomb form of FDN is not mentioned elsewhere in the study. As a result, the work does not support the conclusion brought up in the discussion section.

Authors' Response & Revision:

Many thanks for the reviewer's kind suggestion. The elaborations of the relationship between honeycomb FDN and current study are added, as well as a more detailed description of honeycomb FDN.

The paper mainly studies the method on how to establish a flexibly interconnected and extensible architecture of distribution network based on multi-terminal SOPs, which has laid the foundation for the realisation of the honeycomb FDN [34]. As shown in Fig. 9, the power supply areas are abstractly denoted as a couple of closely packed hexagons, thus representing a primary visualisation of the honeycomb FDN.

With the equipment price of SOP further reducing and the application scope further expanding, the regional distribution network will establish a more compact topology through flexible interconnections. The number of honeycomb cells will increase and the flexible of configurations will be further enhanced. As a result, the characteristics of honeycomb distribution network will gradually come to the fore.

In our prospect, the honeycomb distribution system may be an advanced FDN structure in the future, which enables a more robust grid, by segmenting it into largely autonomous cells. It can be applied to the current grid step by step and may contribute to increase the penetration of renewable energy resources.

The decision maker will focus on the planning of flexible interconnections and expansions from a global perspective, whereas the sources, loads, and energy storage systems are self-organised in each local area.

Fig. 9. Planning results of FDN with different acceptable violation probabilities. The stage to begin flexible reconstruction and the final evolutionary topology of the distribution network differ in terms of the acceptable violation probabilities.

The detailed modifications are shown as follows.

“

The paper mainly studies the method on how to establish a flexibly interconnected and extensible architecture of distribution network based on multi-terminal SOPs, which has laid the foundation for the realisation of the honeycomb FDN [37]. As shown in Fig. 9, the power supply areas are abstractly denoted as a couple of closely packed hexagons, thus representing a primary visualisation of the honeycomb FDN. In our prospect, the honeycomb distribution system may be an advanced FDN structure in the future, which enables a more robust grid, by segmenting it into largely autonomous cells. It can be applied to the current grid step by step and may contribute to increase the penetration of renewable energy resources. The decision maker will focus on the planning of flexible interconnections and expansions from a global perspective, whereas the sources, loads, and energy storage systems are self-organised in each local area.

” (Section 8, page 10, highlighted in blue)

“

[37] Ji, H., Wang, C., Li, P., Zhao, J., Song, G., Ding, F. & Wu, J. An enhanced SOCP-based method for feeder load balancing using the multi-terminal soft open point in active distribution networks. *Appl. Energy* 208, 986-995 (2017)

[38] Zhu, N., Jiang, D., Hu, P. & Yang, Y. Honeycomb active distribution network: a novel structure of distribution network and its stochastic optimization, in 2020 15th IEEE Conference on Industrial Electronics and Applications (ICIEA), 455-462 (2020).

[39] Wang, L., Zu, G., Xu, W., Zhu, W. & Luo, F. Honeycomb distribution networks: concept and central features, in 2022 5th International Conference on Energy, Electrical and Power Engineering (CEEPE), 629-633 (2022).

” (Reference, page 19, highlighted in blue)

Thanks again for your valuable comments!

RESPONSES TO REVIEWER 3'S COMMENTS:

This paper presents a method for planning distribution networks that takes into account multiple resources and allows for long-term allocation strategies. It also addresses uncertainties in source-load relationships to mitigate security risks. However, there are several concerns that need to be addressed by the authors:

Authors' Response & Revision:

We deeply appreciate the reviewer's suggestions. The comments are all valuable and helpful for improving the quality of our paper. Based on the reviewer's comments, the authors made extensive modifications and improvements on the manuscript.

Comment 3.1

The paper does not explicitly mention its contributions. It would be helpful to have a dedicated paragraph for this.

Authors' Response & Revision:

The authors wish to express our sincere gratitude for the insightful comments from the reviewer. We found all the comments and suggestions very helpful for further improving our paper. The authors have carefully addressed all the concerns raised by the reviewer.

The motivation behind this work is to explore and design a novel architecture of distribution networks based on soft open points (SOPs) with the integration of high penetration of distributed generators (DGs) and flexible loads. The paper highlights the flexible regulation and interconnection capabilities of SOPs in spatial dimension, which enables an interconnected and extensible architecture for distribution networks. Conventional distribution networks characterised by the radial and unidirectional structure are expected to evolve into flexible distribution networks (FDNs).

In the FDN, different feeders are flexibly interconnected by SOPs, then the power flow over the entire system can be controlled in closed-loop operation. With the dynamic planning techniques during a long-term period, the converter capacity and terminal number of SOP can expand with the coordinated planning

of sources and loads. Thus, the investment cost can be reasonably assigned to each planning stage, which allows the consistency of FDN planning and avoids investment reset.

Therefore, the specific questions that we aim to address are, how to develop a successive FDN planning strategy over a long duration, and how to determine the siting and sizing of SOPs, electric vehicle charging stations (EVCSs) and photovoltaics (PVs) simultaneously, while considering source-load uncertainties. The planning results can provide comprehensive guidance for future distribution network constructions. The contributions of this paper are summarised and further elaborated as follows.

1) A multi-resource dynamic planning method of FDNs is proposed, in which the configuration of SOPs, PVs and EVCSs is coordinated over a long-term planning period. The flexible reinforcement of FDN can be realised by multiple stages, and advantageous cost-benefits can be achieved compared with traditional planning approach.

2) In the FDN planning model, a probabilistic framework is established to address the strong source-load uncertainties. The security risks are formulated by chance constraints, and the stochastic nonlinear optimisation model is effectively solved based on the modified iterative algorithm. By adjusting the acceptable violation probability in chance constraints, a trade-off between investment efficiency and operational security can be obtained.

The detailed modifications are shown as follows.

“

The contributions of this paper are summarised as follows.

1) A multi-resource dynamic planning method of FDNs is proposed, in which the configuration of SOPs, PVs and EVCSs is coordinated over a long-term planning period. The flexible reinforcement of FDN can be realised by multiple stages, and advantageous cost-benefits can be achieved compared with traditional planning approach.

2) In the FDN planning model, a probabilistic framework is established to address the strong source-load uncertainties. The security risks are formulated by chance constraints, and the stochastic nonlinear optimisation model is effectively solved based on the modified iterative algorithm. By adjusting the acceptable violation probability in chance constraints, a trade-off between investment efficiency and operational security can be obtained.

” (Section 1, page 3, highlighted in blue)

Comment 3.2

The descriptions for cases I, II, and III of planning should be grouped together in the same paragraph.

Authors' Response & Revision:

Many thanks for the reviewer's suggestion. In Section 5, five cases are designed to conduct the cost-benefit analysis. The descriptions for cases are summarized together in the same paragraph, which are given as follows.

Case I: The multi-resource dynamic coordinated planning of FDN is performed.

Case II: The coordinated FDN planning is performed without stage division.

Case III: The energy storage systems (ESSs) planning is performed in the distribution network.

Case IV: The traditional planning method is performed, where the larger-capacity lines and transformers are invested for overloaded feeders.

Case V: The distribution network is not reinforced with the increase of sources and loads.

The detailed modifications are shown as follows.

“

Planning strategy formulation. With the consideration of source-load uncertainties, five cases are designed for FDN planning, and their planning results and cost-benefits are further analysed.

Case I: The multi-resource dynamic coordinated planning of FDN is performed.

Case II: The coordinated FDN planning is performed without stage division.

Case III: The energy storage systems (ESSs) planning is performed in the distribution network.

Case IV: The traditional planning method is performed, where the larger-capacity lines and transformers are invested for overloaded feeders.

Case V: The distribution network is not reinforced with the increase of sources and loads.

” (Section 4, page 5, highlighted in blue)

Comment 3.3

It would be beneficial to investigate the simultaneous use of SOP and line reinforcement for cost reduction in planning by creating another case (case IV) and comparing the results.

Authors' Response & Revision:

The authors would like to appreciate the respectable reviewer for reasonable concerns. The simultaneous consideration of SOP and line reinforcement provides a more comprehensive approach for FDN planning, which exhibits potential economy efficiency. Therefore, in this paper, the simultaneous use of SOP and line reinforcement is considered as Case VI in Discussion section, and the planning results are further compared.

Based on the practical distribution network in case study, the effectiveness of simultaneous use of SOP and line reinforcement is investigated. For illustration, the planning result and cost of Case VI are given in Table 3 and Table 4, respectively.

Table 3 Planning results of the simultaneous use of SOP and line reinforcement

Stage	Reinforcement	SOP allocation: position (capacity /MVA)	EVCS allocation: position (capacity /MVA)	PV allocation: position (capacity /MVA)
I	-	-	49(1.65)	10(2.87) 24(0.45) 39(0.79) 40(0.04) 55(0.79) 64(0.31)
II	-	①7-72 (0.16, 0.04)	28(2.00) 49(2.00) 68(1.95)	10(3.00) 24(1.63) 29(1.92) 39(1.81) 40(0.04) 55(2.54) 60(0.15) 63(0.70) 64(0.55) 76(2.23)
III	Feeder K	①7-72 (0.31, 0.17)	28(2.00) 49(2.00) 68(1.95) 82(2.0) 83(0.91)	10(3.00) 11(3.00) 24(2.78) 29(3.00) 39(2.81) 40(0.04) 55(3.00) 60(1.52) 63(0.74) 64(0.58) 76(2.88)
IV	Feeder K	①7-39-64-72 (1.47, 1.25, 3.15, 3.03)	28(2.00) 49(2.00) 68(2.00) 72(2.00) 82(2.00) 83(1.47)	10(3.00) 11(3.00) 24(3.00) 29(3.00) 39(3.00) 40(2.95) 55(3.00) 60(2.77) 63(3.00) 64(3.00) 76(3.00)

Table 4 Planning cost of the simultaneous use of SOP and line reinforcement

Stage	Investment cost (10 ⁶ CNY)							Operational cost (10 ⁶ CNY)				Sum (10 ⁶ CNY)
	Reinforcement		SOP		EVCS			PV	System loss	Line overload	Voltage violation	
	Line	Transformer	Land exploitation	Converter	Line construction	Land exploitation	Converter	Converter				
I	0.00	0.00	0.00	0.00	0.00	3.00	1.32	4.20	2.36	0.11	0.66	11.64
II	0.00	0.00	2.17	0.07	0.15	4.35	1.61	2.73	1.37	1.15	1.12	14.70
III	1.68	0.53	0.00	0.04	0.00	3.08	0.45	1.66	0.74	0.43	0.57	9.18
IV	0.00	0.00	0.00	0.40	0.51	1.20	0.13	0.45	0.53	0.24	0.00	3.45
Total	2.21		3.35		15.13			9.04	5.00	1.91	2.35	38.98

In the planning scheme, Feeder K is reinforced at Stage III, and a two-terminal SOP is planned in Stage II and evolves into a four-terminal structure in the final stage with the coordination planning with EVCSs and PVs. Compared with Case I, its investment cost (29.72×10^6 CNY) is larger, but its operational cost

(9.26×10^6 CNY) is much smaller. As a result, the total cost of Case VI is reduced by 1.71×10^6 CNY (reduction of 4.20%), which exhibits potential economy efficiency.

The detailed modifications are shown as follows.

“

The simultaneous use of SOP and line reinforcement is further investigated as Case VI. In a feasible planning scheme, Feeder K is reinforced in Stage III, and a two-terminal SOP is planned in Stage II and evolves into a four-terminal structure in the final stage with the coordination planning with EVCSs and PVs. Compared with Case I, its investment cost (29.72×10^6 CNY) is larger, but its operational cost (9.26×10^6 CNY) is much smaller. As a result, the total cost of Case VI is reduced by 1.71×10^6 CNY (reduction of 4.20%), which exhibits potential economy efficiency.

” (Section 8, page 10, highlighted in blue)

Comment 3.4

The paper should provide a general description of the Nataf transformation and a reference.

Authors' Response & Revision:

The authors would like to appreciate the respectable reviewer for reasonable concerns. In this paper, Nataf transformation [38] is used to tackle the correlation between random variables, namely the output of PVs and the demands of conventional loads and EVCSs. The mathematical expressions are supplemented as follows.

Considering a n -dimensional random vector $\xi = (\xi_1, \dots, \xi_n)$ with correlation matrix ρ in an actual probabilistic space, a new random vector $\varsigma = (\varsigma_1, \dots, \varsigma_n)$ in standard normal space and its correlation matrix ρ^ϕ can be obtained by Nataf transformation.

$$\varsigma_i = \Phi^{-1}(G_i(\xi_i)) \quad (2.a)$$

$$\rho_{ij} = \int \int \frac{G_i^{-1}(\Phi(\varsigma_i)) - \mu_i}{\sigma_i} \times \frac{G_j^{-1}(\Phi(\varsigma_j)) - \mu_j}{\sigma_j} \times \phi_2(\varsigma_i, \varsigma_j, \rho_{ij}^\phi) d\varsigma_i d\varsigma_j \quad (2.b)$$

where $G_i(\cdot)$, $G_i^{-1}(\cdot)$, μ_i and σ_i respectively denote the cumulative distribution, inverse cumulative distribution, mean and standard variance of ξ_i . $\Phi(\cdot)$ and $\Phi^{-1}(\cdot)$ respectively denote the cumulative

distribution and inverse cumulative distribution of univariate standard normal distribution. $\phi_2(\cdot)$ denotes the bivariate standard normal distribution. Elements ρ_{ij} and ρ_{ij}^ϕ denote correlation coefficients.

Furthermore, the independent random vector ζ in standard normal space can be obtained.

$$\boldsymbol{\rho}^\phi = \mathbf{L}\mathbf{L}^\mathbf{T} \quad (3.a)$$

$$\boldsymbol{\zeta} = \mathbf{L}^{-1}\boldsymbol{\varsigma} \quad (3.b)$$

where \mathbf{L} denotes the lower triangular matrix by Choleskey decomposition.

The detailed modifications are shown as follows.

“

Nataf transformation. Consider a n -dimensional random vector $\boldsymbol{\xi} = (\xi_1, \dots, \xi_n)$ with correlation matrix $\boldsymbol{\rho}$, where element ρ_{ij} denotes the correlation coefficient between variables ξ_i and ξ_j . With Nataf transformation [38], a new random vector $\boldsymbol{\varsigma} = (\varsigma_1, \dots, \varsigma_n)$ in standard normal space and its correlation matrix $\boldsymbol{\rho}^\phi$ can be obtained.

$$\varsigma_i = \Phi^{-1}(G_i(\xi_i)) \quad (2.a)$$

$$\rho_{ij} = \int \int \frac{G_i^{-1}(\Phi(\varsigma_i)) - \mu_i}{\sigma_i} \times \frac{G_j^{-1}(\Phi(\varsigma_j)) - \mu_j}{\sigma_j} \times \phi_2(\varsigma_i, \varsigma_j, \rho_{ij}^\phi) d\varsigma_i d\varsigma_j \quad (2.b)$$

where $G_i(\cdot)$, $G_i^{-1}(\cdot)$, μ_i and σ_i respectively denote the cumulative distribution, inverse cumulative distribution, mean and standard variance of ξ_i . $\Phi(\cdot)$ and $\Phi^{-1}(\cdot)$ respectively denote the cumulative distribution and inverse cumulative distribution of univariate standard normal distribution. $\phi_2(\cdot)$ denotes the bivariate standard normal distribution. Element ρ_{ij}^ϕ denotes the correlation coefficient between variables ς_i and ς_j .

Furthermore, the independent random vector ζ in standard normal space can be obtained based on Choleskey decomposition.

$$\boldsymbol{\rho}^\phi = \mathbf{L}\mathbf{L}^\mathbf{T} \quad (3.a)$$

$$\boldsymbol{\zeta} = \mathbf{L}^{-1}\boldsymbol{\varsigma} \quad (3.b)$$

where \mathbf{L} denotes the lower triangular matrix.

” (Section 9, page 11, highlighted in blue)

“

[38] Chen, Y., Wen, J. & Cheng, S. Probabilistic load flow method based on Nataf transformation and Latin hypercube sampling. IEEE Trans. on Sustain. Energy 4(2), 294-301 (2012)

” (Reference, page 19, highlighted in blue)

Comment 3.5

In figure 8.a, the violation risk of the lower bound for voltages is shown, but it is unclear what exactly is being displayed. A description is needed to clarify this. The same applies to figures 8.b and 8.c.

Authors’ Response & Revision:

Many thanks for the reviewer’s kind suggestion. Fig .8 shows that FDN violation risks gradually converge to a predefined level during iterations. In particularly, subfigure (a) shows the violation risk of the lower bound for voltages, subfigure (b) shows the violation risk of the upper bound for voltages and subfigure (c) shows the violation risk of the upper bound for currents. The detailed description is added for clarifications.

Fig. 8. Violation risks of FDN during iterations. FDN planning strategy is updated iteratively with the corrected margins of security constraints, thus ensuring the violation risks converge to the predefined level. Convergence efficiency is improved using the modified iterative algorithm. The x-axis ticks represent the indices of stage. At each stage interval (divided by grey dashed lines), the violation risks of nodal voltages or line currents are illustrated, with labels (A, B, ..., K) at the top indicating corresponding feeder names. The y-axis ticks represent the violation risks. The initial violation risks (blue lines) are rapidly reduced to the vicinity of 5% (dark red dashed line) only after one iteration (orange lines). Then, slight reductions of violation risks are conducted during later iterations. The iteration stops (red lines) when all violation risks are controlled below the acceptable probability. (a) Violation risk of the lower bound for voltages. (b) Violation risk of the upper bound for voltages. (c) Violation risk of the upper bound for currents.

The detailed modifications are shown as follows.

“

Fig. 8. Violation risks of FDN during iterations. FDN planning strategy is updated iteratively with the corrected margins of security constraints, thus ensuring the violation risks converge to the predefined level. Convergence efficiency is improved using the modified iterative algorithm. The x-axis ticks represent the indices of stage. At each stage interval (divided by grey dashed lines), the violation risks of nodal voltages or line currents are illustrated, with labels (A, B, ..., K) at the top indicating corresponding feeder names. The y-axis ticks represent the violation risks. The initial violation risks (blue lines) are rapidly reduced to the vicinity of 5% (dark red dashed line) only after one iteration (orange lines). Then, slight reductions of violation risks are conducted during later iterations. The iteration stops (red lines) when all violation risks are controlled below the acceptable probability. (a) Violation risk of the lower bound for voltages. (b) Violation risk of the upper bound for voltages. (c) Violation risk of the upper bound for currents.

” (Section 6, page 8, highlighted in blue)

Comment 3.6

The 1 percent loss coefficient for SOP seems highly optimistic and its impact on the system's total loss should be considered.

Authors' Response & Revision:

We really appreciate the reviewer's comments. Considering that the efficiency of converter has reached more than 98% [30], the loss factor of SOP converter is set to 0.02 [31]. The SOP converter loss has been involved in the system operational cost along with the network loss.

$$\phi_u^{\text{OP}} = 8760 \cdot (\phi_u^{\text{NET,LS}} + \phi_u^{\text{SOP,LS}}) \quad (9.a)$$

$$\phi_u^{\text{NET,LS}} = c_u^{\text{SL}} \sum_{ij \in \Omega_b} R_{ij} I_{ij,u}^2 \quad (9.b)$$

$$\phi_u^{\text{SOP,LS}} = c_u^{\text{SL}} \sum_{k \in \mathcal{L}} \sum_{i \in \Omega_k} P_{i,k,u}^{\text{SOP,LS}} \quad (9.c)$$

where $\phi_u^{\text{NET,LS}}$ and $\phi_u^{\text{SOP,LS}}$ denote the network loss and SOP converter loss cost, respectively. c_u^{SL} denote the price of power loss in stage u , which is generally assigned as electricity price. R_{ij} denotes

the resistance of branch ij . $I_{ij,u}$ denotes the current magnitude of branch ij in stage u . $P_{i,k,u}^{\text{SOP,LOS}}$ denotes the active power losses at node i in scheme k in stage u .

The detailed modifications are shown as follows.

“

Considering that the efficiency of converter has reached more than 98% [30], the loss factor of SOP converter is set to 0.02 [31].

” (Section 3, page 4, highlighted in blue)

“

[30] Zhang, S., Fang, Y., Zhang, H., Cheng, H. & Wang, X. Maximum hosting capacity of photovoltaic generation in sop-based power distribution network integrated with electric vehicles. IEEE Trans. Industr. Inform. 18(11), 8213-8224 (2022)

[31] Wang, C., Song, G., Li, P., Ji, H., Zhao, J. & Wu, J. Optimal siting and sizing of soft open points in active electrical distribution networks. Appl. Energy 189, 301-309 (2017)

” (Reference, page 19, highlighted in blue)

Comment 3.7

The paper should explain the logic behind the location designation for SOP and whether SOP capacity is involved in these schemes. Additionally, the step size for SOP capacity in the schemes should be addressed.

Authors' Response & Revision:

Many thanks for the reviewer's suggestion. The siting designation for SOP is elaborated as follows. First, a set of available nodes is determined for SOP connections. In this paper, the terminal nodes of existing tie lines are selected as the available nodes. Second, the topologies of SOP planning schemes are generated without exceeding the maximum number of SOP terminals, and the length of the line to be reconstructed in each scheme is calculated.

The evolution of SOP planning schemes is formulated as follows. Eq. (6.a) represents the formation of the SOP planning scheme set. Eqs. (6.a)-(6.d) respectively instantiate the set of SOP planning schemes with $\tau = 2,3,4$. Eq. (6.e) represents the set of SOP planning schemes that contain the same terminal node.

$$\mathcal{L} = \cup_{k=1}^{N_f} \mathcal{L}(k) = \cup_{k=1}^{N_f} \cup_{\tau=2}^{M_f} \mathcal{L}(k, \tau) \quad (6.a)$$

$$\mathcal{L}(k, 2) = \{k | \text{crad}(\Omega_k) = 2\} \quad (6.b)$$

$$\mathcal{L}(k, 3) = \{k' | \text{crad}(\Omega_{k'}) = 3, \Omega_{k'} \supseteq \Omega_k\} \quad (6.c)$$

$$\mathcal{L}(k, 4) = \{k'' | \text{crad}(\Omega_{k''}) = 4, \Omega_{k''} \supseteq \Omega_k\} \quad (6.d)$$

$$\mathcal{L}_i = \{k | \Omega_k \ni i, \forall i \in \Omega_s\} \quad (6.e)$$

where Ω_s denotes the set of available nodes. k denotes scheme index. Ω_k denotes the set of nodes in scheme k . N_f denotes the number of schemes. M_f denotes the maximum number of SOP terminals. \mathcal{L} denotes the set of total schemes. \mathcal{L}_i denotes the set of SOP planning schemes containing node i . $\mathcal{L}(k)$ denote the set of schemes evolved from scheme k , and $\mathcal{L}(k, \tau)$ denotes the τ -terminal SOP planning schemes in set $\mathcal{L}(k)$. $\text{crad}(\cdot)$ denotes the cardinality of a set.

The specific capacity of SOP converter is not predefined for each planning scheme, which is optimised in the proposed planning model within a maximum capacity, as formulated in Eq. (10).

$$\sum_{k \in \mathcal{L}_i} \alpha_{k,u} \leq 1 \quad (10.a)$$

$$\alpha_{k,u-1} \leq \sum_{k' \in \mathcal{L}(k)} \alpha_{k',u} \quad (10.b)$$

$$\sum_{k \in \mathcal{L}_i} S_{i,k,u-1}^{\text{SOP}} \leq \sum_{k \in \mathcal{L}_i} S_{i,k,u}^{\text{SOP}} \quad (10.c)$$

$$S_{k,u}^{\text{SOP}} = \sum_{i \in \Omega_k} S_{i,k,u}^{\text{SOP}} \quad (10.d)$$

$$\pi \alpha_{k,u} \leq S_{k,u}^{\text{SOP}} \leq S_k^{\text{SOP,max}} \alpha_{k,u} \quad (10.e)$$

where $\alpha_{k,u}$ is a binary variable, indicating whether SOP planning scheme k is selected in stage u . $S_{k,u}^{\text{SOP}}$ denotes the converter capacity of SOP planning scheme k in stage u . $S_{i,k,u}^{\text{SOP}}$ denotes the converter capacity at node i in scheme k in stage u . $S_k^{\text{SOP,max}}$ denotes the maximum capacity of SOP in scheme k . π denotes a minimal positive value.

In the proposed FDN planning model, the converter capacity of SOP is formulated as continuous variables for effective solutions. However, considering the modularisation requirements, the converter capacity of SOP can only be an integral multiple of a unit module. Thus, the indeed installed SOP capacity is determined by rounding the corresponding variables in the solution. In this paper, the unit module capacity for SOP converter is set to 10 kVA.

The detailed modifications are shown as follows.

“

The maximum converter capacity of a single SOP is 10 MVA, and the unit module capacity is set to 10 kVA.

” (Section 3, page 4, highlighted in blue)

“

Evolution of FDN planning. The number of terminals, and the siting and sizing of SOP can be flexibly designed in each stage. First, a set of available nodes is determined for SOP connections. In this paper, the terminal nodes of existing tie lines are selected as the available nodes. Second, the topologies of SOP planning schemes are generated without exceeding the maximum number of SOP terminals, and the length of the line to be reconstructed in each scheme is calculated.

” (Section 9, page 12, highlighted in blue)

Comment 3.8

There are many symbols that are not introduced or introduced in later equations. For example, equations 5, y , and Ωy are not introduced, and equations 6 and $\beta_{i,u}$ are not explained. A review and correction of these mistakes is necessary, and a nomenclature table may be useful.

Authors' Response & Revision:

The authors would like to thank the reviewer for carefully perusing our paper. The explanations of variables y and Ωy in equations (5), and $\beta_{i,u}$ in equations (6) are supplemented. Furthermore, similar mistakes are checked throughout the entire paper. Remark: Equations (5) are renamed as (7), and equations (6) are renamed as (8).

The detailed modifications are shown as follows.

“

$$f = \min \sum_{u \in \Omega_U} \sum_{y \in \Omega_Y} \lambda_{yu} (\varepsilon \phi_u^{\text{CO}} + \phi_u^{\text{OP}}) \quad (7.a)$$

where Ω_U and Ω_Y denote the set of stages and years, respectively. Y denotes the duration of each planning stage. y denotes the index of years in each planning stage.

” (Section 9, page 12, highlighted in blue)

“

$$\phi_u^{\text{EVCS,ST}} = c_u^{\text{ST}} (\sum_{i \in \Omega_e} \beta_{i,u} - \sum_{i \in \Omega_e} \beta_{i,u-1}) \quad (8.e)$$

where $\beta_{i,u}$ is a binary variable, indicating whether the EVCS is constructed at node i in stage u .

” (Section 9, page 13, highlighted in blue)

Comment 3.9

The paper does not include the power flow model and related equations. At least a brief explanation of the model and power flow variables is required.

Authors' Response & Revision:

The authors would like to thank the reviewer for the valuable suggestion. We have provided the power flow constraints of distribution network based on DistFlow branch model [40], as shown in (13). The DistFlow branch model describes the power flow mechanism precisely and has been applied widely in distribution networks [41].

Constraints (13.a) and (13.b) represent the active and reactive power balance of node i . Ohm's law over branch ij is expressed as (13.c). The current magnitude of each line can be determined by (13.d). Constraints (13.e) and (13.f) indicate the total active and reactive power injection of node i .

$$\sum_{ij \in \Omega_b} (P_{ij,u} - R_{ij} I_{ij,u}^2) + P_{j,u} = \sum_{jr \in \Omega_b} P_{jr,u} \quad (13.a)$$

$$\sum_{ij \in \Omega_b} (Q_{ij,u} - X_{ij} I_{ij,u}^2) + Q_{j,u} = \sum_{jr \in \Omega_b} Q_{jr,u} \quad (13.b)$$

$$V_{i,u}^2 - V_{j,u}^2 + (R_{ij}^2 + X_{ij}^2) I_{ij,u}^2 - 2(R_{ij} P_{ij,u} + X_{ij} Q_{ij,u}) = 0 \quad (13.c)$$

$$I_{ij,u}^2 V_{i,u}^2 - (P_{ij,u}^2 + Q_{ij,u}^2) = 0 \quad (13.d)$$

$$P_{i,u} = P_{i,u}^S + P_{i,u}^{PV} + \sum_{k \in \mathcal{L}} P_{i,k,u}^{SOP} - P_{i,u}^{LD} - P_{i,u}^{EV} \quad (13.e)$$

$$Q_{i,u} = Q_{i,u}^S + Q_{i,u}^{PV} + \sum_{k \in \mathcal{L}} Q_{i,k,u}^{SOP} - Q_{i,u}^{LD} \quad (13.f)$$

where $P_{ij,u}$ and $Q_{ij,u}$ denote the active and reactive power flow of branch ij in stage u , respectively. R_{ij} and X_{ij} denote the resistance and reactance of branch ij , respectively. $I_{ij,u}$ denotes the current magnitude of branch ij in stage u . $V_{i,u}$ denotes the voltage magnitude of node i in stage u .

Note that the proposed FDN planning problem is a mixed-integer nonlinear programming (MINLP) model. A convex relaxation [42] is adopted to convert the MINLP model to a mixed-integer second-order conic programming (MISOCP) formulation, which can be efficiently computed by commercial solvers. The convex relaxation of power flow constraints is mathematically described as follows.

$$\sum_{ij \in \Omega_b} (P_{ij,u} - R_{ij} l_{ij,u}) + P_{j,u} = \sum_{jr \in \Omega_b} P_{jr,u} \quad (14.a)$$

$$\sum_{ij \in \Omega_b} (Q_{ij,u} - X_{ij} l_{ij,u}) + Q_{j,u} = \sum_{jr \in \Omega_b} Q_{jr,u} \quad (14.b)$$

$$v_{i,u} - v_{j,u} + (R_{ij}^2 + X_{ij}^2) l_{ij,u} - 2(R_{ij} P_{ij,u} + X_{ij} Q_{ij,u}) = 0 \quad (14.c)$$

$$\left\| \begin{array}{c} 2P_{ij,u} \\ 2Q_{ij,u} \\ l_{ij,u} - v_{i,u} \end{array} \right\|_2 \leq l_{ij,u} + v_{i,u} \quad (14.d)$$

where $l_{ij,u}$ denotes the square of current magnitude of branch ij in stage u . $v_{i,u}$ denotes the square of voltage magnitude of node i in stage u . Namely, $l_{ij,u} = I_{ij,u}^2$ and $v_{i,u} = V_{i,u}^2$.

The detailed modifications are shown as follows.

“

Constraints of FDN operation. The power flow constraints of distribution network are formulated based on DistFlow branch model [40], which describes the power flow mechanism precisely and has been applied widely in distribution networks [41].

$$\sum_{ij \in \Omega_b} (P_{ij,u} - R_{ij} I_{ij,u}^2) + P_{j,u} = \sum_{jr \in \Omega_b} P_{jr,u} \quad (13.a)$$

$$\sum_{ij \in \Omega_b} (Q_{ij,u} - X_{ij} I_{ij,u}^2) + Q_{j,u} = \sum_{jr \in \Omega_b} Q_{jr,u} \quad (13.b)$$

$$V_{i,u}^2 - V_{j,u}^2 + (R_{ij}^2 + X_{ij}^2) I_{ij,u}^2 - 2(R_{ij} P_{ij,u} + X_{ij} Q_{ij,u}) = 0 \quad (13.c)$$

$$I_{ij,u}^2 V_{i,u}^2 - (P_{ij,u}^2 + Q_{ij,u}^2) = 0 \quad (13.d)$$

$$P_{i,u} = P_{i,u}^S + P_{i,u}^{PV} + \sum_{k \in \mathcal{L}} P_{i,k,u}^{SOP} - P_{i,u}^{LD} - P_{i,u}^{EV} \quad (13.e)$$

$$Q_{i,u} = Q_{i,u}^S + Q_{i,u}^{PV} + \sum_{k \in \mathcal{L}} Q_{i,k,u}^{SOP} - Q_{i,u}^{LD} \quad (13.f)$$

where Ω_b denotes the set of branches. $P_{ij,u}$ and $Q_{ij,u}$ denote the active and reactive power flow of branch ij in stage u , respectively. R_{ij} and X_{ij} denote the resistance and reactance of branch ij , respectively. $V_{i,u}$ denotes the voltage magnitude of node i in stage u . $P_{i,u}$ and $Q_{i,u}$ denote the total active and reactive power injection at node i in stage u , respectively. $P_{i,u}^S$, $Q_{i,u}^S$, $P_{i,u}^{PV}$, $Q_{i,u}^{PV}$, $P_{i,u}^{LD}$ and $Q_{i,u}^{LD}$ denote the active and reactive power injection by substation, PV and load at node i in stage u , respectively. $P_{i,k,u}^{SOP}$ and $Q_{i,k,u}^{SOP}$ denote the active and reactive power injection by SOP at node i in scheme k in stage u , respectively. $P_{i,u}^{EV}$ denote the active power injection by EV at node i in stage u .

Note that the proposed FDN planning problem is a mixed-integer nonlinear programming (MINLP) model. A convex relaxation [42] is adopted to convert the MINLP model to a mixed-integer second-order conic programming (MISOCP) formulation, which can be efficiently computed by commercial solvers. The convex relaxation of power flow constraints is mathematically described as follows.

$$\sum_{ij \in \Omega_b} (P_{ij,u} - R_{ij} l_{ij,u}) + P_{j,u} = \sum_{jr \in \Omega_b} P_{jr,u} \quad (14.a)$$

$$\sum_{ij \in \Omega_b} (Q_{ij,u} - X_{ij} l_{ij,u}) + Q_{j,u} = \sum_{jr \in \Omega_b} Q_{jr,u} \quad (14.b)$$

$$v_{i,u} - v_{j,u} + (R_{ij}^2 + X_{ij}^2) l_{ij,u} - 2(R_{ij} P_{ij,u} + X_{ij} Q_{ij,u}) = 0 \quad (14.c)$$

$$\left\| \begin{array}{c} 2P_{ij,u} \\ 2Q_{ij,u} \\ l_{ij,u} - v_{i,u} \end{array} \right\|_2 \leq l_{ij,u} + v_{i,u} \quad (14.d)$$

where $l_{ij,u}$ denotes the square of current magnitude of branch ij in stage u . $v_{i,u}$ denotes the square of voltage magnitude of node i in stage u . Namely, $l_{ij,u} = I_{ij,u}^2$ and $v_{i,u} = V_{i,u}^2$.

” (Section 9, page 14, highlighted in blue)

“

[40] Baran, M.E. & Wu, F.F. Optimal capacitor placement on radial distribution systems. IEEE Trans. Power Deliv. 4(1), 725–734 (1989)

[41] Daniel, K.M., Florian, D., Henrik, S., Steven, H.L., Sambuddha, C., Ross, B. & Javad, L. A survey of distributed optimization and control algorithms for electric power systems. IEEE Trans. Smart Grid 8(6), 2941–2962 (2017)

[42] Lavaei, J. & Low, S.H. Zero duality gap in optimal power flow problem. IEEE Trans. Power Syst. 27(1), 92–107 (2012)

” (Reference, page 20, highlighted in blue)

Comment 3.10

More explanation is needed for the method used to obtain equations 13, as it involves a large number of power flow calculations.

Authors' Response & Revision:

Many thanks for the careful review from the reviewer. The Eq. (13) is renamed as Eq. (19) and is reformulated for rationality improvements, which are shown as follows. The operational cost is calculated for evaluating the operational economy when the planning methods are adopted in practice.

$$\phi_u^{\text{OP,all}} = 8760 \cdot (\phi_u^{\text{FDN,LS}} + \phi_u^{\text{VOLT,VL}} + \phi_u^{\text{CURT,VL}}) \quad (19.a)$$

$$\phi_u^{\text{FDN,LS}} = c_u^{\text{SL}} \mathbb{E}[\sum_{ij \in \Omega_b} R_{ij} l_{ij,u} + \sum_{k \in \mathcal{L}} \sum_{i \in \Omega_k} P_{i,k,u}^{\text{SOP,LS}}] \quad (19.b)$$

$$\phi_u^{\text{VOLT,VL}} = c_u^{\text{VL}} \mathbb{E} \left[\sum_{i \in \Omega_n} P_{i,u}^{\text{LD}} : v_{i,u} < V_{\min}^2 \mid \mid v_{i,u} > V_{\max}^2 \right] \quad (19.c)$$

$$\phi_u^{\text{CURT,VL}} = c_u^{\text{VL}} \mathbb{E} \left[\sum_{ij \in \Omega_b} \sum_{r \in \Omega_{n,ij}} P_{r,u}^{\text{LD}} : l_{ij,u} > I_{\max}^2 \right] \quad (19.d)$$

where $\phi_u^{\text{FDN,LS}}$ denotes the cost of FDN loss. $\phi_u^{\text{VOLT,VL}}$ and $\phi_u^{\text{CURT,VL}}$ denote the costs of voltage violation and line overload, respectively. Ω_n denotes the set of nodes, and $\Omega_{n,ij}$ denotes the set of the nodes downstream of branch ij . c_u^{VL} denotes the penalty price.

As for the involvement of a large number of power flow calculations, the detailed interpretations are given as follows. First, a sample set of source-load profiles as input is generated from the established probability distribution functions. Next, based on the determined planning strategies of SOPs, PVs and EVCSs in the FDN, a number of power flow calculations is executed by Monte Carlo simulations, which is sound for evaluations. If more computational efficiency is needed, low-rank approximation can be adopted as an alternative. Finally, the average of network loss and load loss are used to quantify the operational cost.

The detailed modifications are shown as follows.

“

Revised operational cost. After the planning strategy of FDN is determined, a large number of power flow calculations based on Monte Carlo method is executed to compute the revised operational cost.

The penalty cost is attributed to the load loss affected by potential security risks. Especially, the penalty cost for voltage violation is computed as the sum of the active power of the loads on the nodes where voltages exceed the safe range. The penalty cost for line overloading is computed as the sum of the active power of the loads located at the downstream nodes of the overload line. The calculation method is formulated as follows.

$$\phi_u^{\text{OP,all}} = 8760 \cdot (\phi_u^{\text{FDN,LS}} + \phi_u^{\text{VOLT,VL}} + \phi_u^{\text{CURT,VL}}) \quad (19.a)$$

$$\phi_u^{\text{FDN,LS}} = c_u^{\text{SL}} \mathbb{E} \left[\sum_{ij \in \Omega_b} R_{ij} l_{ij,u} + \sum_{k \in \mathcal{L}} \sum_{i \in \Omega_k} P_{i,k,u}^{\text{SOP,LS}} \right] \quad (19.b)$$

$$\phi_u^{\text{VOLT,VL}} = c_u^{\text{VL}} \mathbb{E} \left[\sum_{i \in \Omega_n} P_{i,u}^{\text{LD}} : v_{i,u} < V_{\min}^2 \mid \mid v_{i,u} > V_{\max}^2 \right] \quad (19.c)$$

$$\phi_u^{\text{CURT,VL}} = c_u^{\text{VL}} \mathbb{E} \left[\sum_{ij \in \Omega_b} \sum_{r \in \Omega_{n,ij}} P_{r,u}^{\text{LD}} : l_{ij,u} > I_{\max}^2 \right] \quad (19.d)$$

where $\phi_u^{\text{FDN,LS}}$ denotes the cost of FDN loss. $\phi_u^{\text{VOLT,VL}}$ and $\phi_u^{\text{CURT,VL}}$ denote the costs of voltage violation and line overload, respectively. Ω_n denotes the set of nodes, and $\Omega_{n,ij}$ denotes the set of the nodes downstream of branch ij . c_u^{VL} denotes the penalty price, which is generally assigned as the electricity price.

Additionally, the above penalty costs are not included in the objective of planning model, for the reason that the nodal voltages and line currents are restricted within a safe range in Eq. (18). However, the chance constrains allow violations within a permitted probability, so the costs of voltage violation and line overload are involved in the economic estimation of the planning scheme.

” (Section 9, page 15, highlighted in blue)

Comment 3.11

It may be better to move the solution procedure section to the beginning of the method part so that readers can have a general understanding of the solving procedure.

Authors' Response & Revision:

Many thanks for the reviewer's suggestion. The “Solution procedure” part and is moved at the beginning of the Methods section. In addition, to make the correspondence clearer, the equation indices are attached to the models which are used in each step.

The detailed modifications are shown as follows.

“

Solution procedure. The detailed procedure for solving the FDN planning model is given as follows.

- 1) Input the distribution network parameters, and determine the random variables and correlation matrix;
- 2) Quantify the source-load uncertainties based on Gaussian mixture model in Eq. (1) and generate samples based on Nataf transformation in Eq. (2);
- 3) Set iteration counter $k=0$;

4) Check whether k is less than or equal to k_{\max} . If satisfied, continue to Step 5); otherwise, proceed to Step 9);

5) Solve the deterministic planning model in Eq. (25) to obtain the planning strategy, including the optimised allocation of SOPs, EVCSs and PVs;

6) Execute uncertainty propagation to obtain the probabilistic characteristics of nodal voltages and line currents;

7) Check whether all chance constraints satisfy the predefined risk level. If satisfied, proceed to Step 9); otherwise, continue to Step 8);

8) Update the bounds of the security constraints; update $k=k+1$ and proceed to Step 4);

9) Record the solved planning strategy and calculate the total cost.

” (Section 9, page 11, highlighted in blue)

Comment 3.12

The paper does not mention the type of optimization problem or the solver used. If linearizations or relaxations have been implemented, they should be mentioned.

Authors' Response & Revision:

Many thanks for the reviewer's comment. The power flow constraints for distribution networks are formulated based on DistFlow branch model. Note that the proposed optimization problem for FDN planning is originally a mixed-integer nonlinear programming (MINLP) model.

For effective solutions, a convex relaxation is implemented. First, variable substitution is used to realize linearization. Then, Eq. (13c) is further relaxed to a second-order cone constraint. Consequently, the original MINLP model converted into a mixed-integer second-order conic programming (MISOCP) formulation, which can be efficiently computed by commercial solvers. In this paper, Gurobi is adopted for model solutions. Equations (13) and (14) are given in Response 3.9. In addition, to evaluate the accuracy of the convex relaxation, an index is defined to quantify the relaxation deviation between the original non-convex problem and the convexified problem.

Further considering the uncertainties originating from distributed sources and loads, the security risks of FDN are formulated by chance constraints. As a result, the optimisation model is eventually extended

to a chance-constrained MISOCP formulation. Using the proposed modified iterative algorithm, the chance constraints can be tractably solved.

The detailed modifications are shown as follows.

“

To evaluate the accuracy of the convex relaxation for the proposed model, an index [43] is defined to quantify the relaxation deviation as follows.

$$J_u = \|l_{ij,u}v_{i,u} - P_{ij,u}^2 - Q_{ij,u}^2\|_\infty \quad (15)$$

where J_u denotes the index to evaluate the relaxation deviation, indicating whether the SOCP-relaxed optimal solution is accurate or not. If the gap is smaller than a pre-specified tolerance, the optimal solution is accepted as exact.

” (Section 9, page 15, highlighted in blue)

“

[43] Wei, W., Wang, J., Li, N. & Mei, S. Optimal Power flow of radial networks and its variations: a sequential convex optimization approach. IEEE Trans. Smart Grid 8(6), 2974-2987 (2017)

” (Reference, page 20, highlighted in blue)

Thanks again for your valuable comments!

REVIEWER COMMENTS

Reviewer #1 (Remarks to the Author):

The revised paper is an improvement on the first submission, with clearer novelty and more comprehensive results. My comments from the first submission have been thoroughly addressed, but some of the new text has some English language issues, for example on page 2 line 66 mandatory is used instead of mandated, and on line 68 the word piles is used.

I suggest a thorough final English language review prior to publication.

Reviewer #2 (Remarks to the Author):

The paper has undergone significant enhancements, with many of my comments addressed. However, one particular point remains unresolved:

Concerning the paragraph discussing the coordinated planning of SOPs, EVCSs, and PVs to optimize overall social benefits, the paper offers extensive planning guidance for power companies, energy suppliers, and users in distribution networks.

My query pertains to the planning's perspective: "Does the coordinated planning of SOPs, EVCSs, and PVs, as outlined in this paper, fail to identify the Pareto optimal solution, thereby neglecting the interests of multiple parties in practical operations? Whose perspective does the planning problem aim to address?"

Reviewer #3 (Remarks to the Author):

The authors have effectively responded to the issues raised by this reviewer, and I do not have any additional inquiries.

RESPONSES TO REVIEWER 1'S COMMENTS:

The revised paper is an improvement on the first submission, with clearer novelty and more comprehensive results. My comments from the first submission have been thoroughly addressed, but some of the new text has some English language issues, for example on page 2 line 66 mandatory is used instead of mandated, and on line 68 the word piles is used. I suggest a thorough final English language review prior to publication.

Authors' Response & Revision:

The authors would like to thank the reviewer for the valuable comments for further improving the paper's quality. The writing and language are carefully checked and polished in the revised manuscript, and a thorough proof read is performed to make the paper clearer.

Many thanks for your kind suggestions!

RESPONSES TO REVIEWER 2’S COMMENTS:

The paper has undergone significant enhancements, with many of my comments addressed. However, one particular point remains unresolved:

Concerning the paragraph discussing the coordinated planning of SOPs, EVCSs, and PVs to optimize overall social benefits, the paper offers extensive planning guidance for power companies, energy suppliers, and users in distribution networks.

My query pertains to the planning's perspective: "Does the coordinated planning of SOPs, EVCSs, and PVs, as outlined in this paper, fail to identify the Pareto optimal solution, thereby neglecting the interests of multiple parties in practical operations? Whose perspective does the planning problem aim to address?"

Authors’ Response & Revision:

The authors would like to appreciate the respectable reviewer for the reasonable concern. The planning perspective and motivation are further clarified. The orientation of planning results is illustrated, while the handling of multi-objective optimization is elaborated and discussed. We sincerely hope that these modifications can make our claim clearer and more intelligible.

1) Planning perspective

The planning of flexible distribution network (FDN) is conducted from the perspective of power companies. The power companies aim to optimize the siting and sizing of soft open points (SOPs) for the distribution network to improve its hosting capability of electric vehicle charging stations (EVCSs) and photovoltaics (PVs).

2) Motivation of the coordinated planning

The power companies undertake the investment of SOPs, which belong to the assets of distribution networks. The increasing EVCSs and PVs will change the power flow of distribution networks, which affects the location and capacity of SOPs. Thus, the power companies have the motivation to perform a coordinated planning of SOPs, EVCSs and PVs, and the FDN planning model should accommodate the allocation strategy of EVCSs and PVs.

Furthermore, the proposed dynamic planning method enables the configuration of SOPs to adapt to the changes in source and load by stages. The expected capacities and available positions of household PVs

and EVCSs in each stage can be estimated, and incorporated by the FDN planning model. With the simultaneous consideration of grid, source and load in a multi-stage planning framework, the planning guidance exhibits high compatibility. As a result, the total investment for distribution network reinforcement can be reduced or delayed.

3) Orientation of planning results

The power companies can participate in the construction of EVCSs and PVs based on the coordinated planning results. On the other hand, as the EVCSs and PVs in distribution networks are invested and built by public stakeholders, the planning strategy of EVCSs and PVs is provided as guidance and suggestion for them. The EVCSs and PVs constructed at the suggested locations will be better regulated and supported by SOPs to ensure the secure hosting of renewable generation and charging load. Therefore, it is beneficial to formulate planning guidance for individual users, especially for the energy suppliers that make profits by providing public generating and charging services.

At the same time, note that the energy suppliers and users are not mandated to follow the planning result of sources and loads. But the user-built EVCSs and PVs may bring operational violations to the distribution network, resulting in the inability to plug themselves into the grid. Under such conditions, users have to pay additional costs to dispatch energy storage systems or other controllable resources for regulations, while wasting the public services provided by configured SOPs.

To further elaborate the effectiveness of the proposed coordinated planning method, a new case is investigated as Case VI, and its description is given as follows.

Case I: The multi-resource dynamic coordinated planning of FDN is performed.

Case VI: The power companies conduct SOP planning without the consideration of increasing EVCSs and PVs, while the allocation of EVCSs and PVs is determined and invested by users or energy suppliers.

Based on the practical distribution network in case study, the siting and sizing of SOP in Case VI are solved and illustrated in Table 1. The result indicates that a two-terminal SOP is built in Stage I-II, and is extended to a multi-terminal structure in Stage III-IV. However, the number and capacity of SOP in Stage III-IV are less than those in Case I, due to neglecting the effect of EVCSs and PVs integration.

Table I Planning results of SOPs

		Stage I	Stage II	Stage III	Stage IV
SOP allocation:	Case I	--	①7-72(0.16, 0.04)	①7-72-83(0.81, 0.64, 0.79)	①7-64-72-83(0.81, 3.34, 1.06, 1.76)
position					②29-39(0.96, 0.97)
(capacity /MVA)	Case VI	①7-72(0.49, 0.00)	①7-72(0.74, 0.00)	①7-72-83(0.90, 0.00, 0.01)	①7-39-72-83(0.99, 0.24, 0.09, 0.09)

In Case VI, the allocation of EVCSs and PVs is stochastically generated for simulations, and the total integration capacity of EVCSs and PVs in each stage is consistent with that in Case I. The adapted allocation scenario of EVCSs and PVs is given in Table 2.

Table 2 The allocation of EVCSs and PVs in Case VI

	Stage I	Stage II	Stage III	Stage IV
EVCS allocation:	82(0.76)	72(2.06) 82(1.06) 83(2.85)	28(0.91) 49(0.45) 68(0.09)	28(0.98) 49(0.79) 68(0.66)
position (capacity /MVA)	83(0.90)		72(2.95) 82(1.26) 83(3.23)	72(3.46) 82(1.78) 83(3.82)
PV allocation:	24(2.59)	24(2.86) 39(2.21)	10(1.62) 11(1.73) 24(3.19)	10(1.88) 11(2.47) 24(4.25)
position (capacity /MVA)	40(2.60)	40(2.84) 55(1.49)	29(1.58) 39(3.10) 40(3.10)	29(1.63) 39(4.86) 40(4.58)
	64(0.02)	63(2.03) 64(1.13)	55(2.32) 60(0.88) 63(3.39)	55(3.99) 60(1.34) 63(3.41)
			64(2.45)	64(3.11) 76(1.24)

Then, the violation risks of the distribution network over the whole planning period in Case VI are calculated and illustrated in Table 3. In Case I, the probabilities of voltage violation and line overload in the distribution network are less than 5%. In Case VI, the risks of voltage violation in Stage II-IV exceed 15%, while the risks of line overload exceed 25%. The notable risks will challenge the secure operation of distribution networks, and hinder EVCSs and PVs integration into the grid.

Table 3 Violation risks of the distribution network

Stage	Maximum violation risk of the lower bound for voltages		Maximum violation risk of the upper bound for voltages		Maximum violation risk of the upper bound for line load rates	
	Case I	Case VI	Case I	Case VI	Case I	Case VI
I	3.675%	4.625%	0.000%	0.000%	0.120%	3.910%
II	4.895%	26.465%	0.000%	15.255%	1.250%	26.025%
III	2.000%	35.955%	0.000%	26.150%	4.355%	35.585%
IV	2.420%	39.000%	2.930%	41.330%	4.895%	41.765%

Furthermore, the cost benefits are compared between Case I and Case VI, which are given in Table 4. Because of ignoring the hosting demand of EVCSs and PVs, the SOP investment cost in Case VI is reduced by 9.17% compared with Case I. However, the operational cost for violations increases by 4.70

times, and the total cost increases by 3.31 times. The result indicates that if FDN planning is not performed in a coordinated framework, the regulation ability of SOPs cannot be fully utilized, and the randomness caused by PVs and loads will exacerbate the operation of distribution network. The separate decision-making leads to worse economic outcomes for the overall benefit of society.

Table 4 Cost-benefit analysis of the distribution network

Stage	SOP investment cost (10 ⁶ CNY)		Operational cost (10 ⁶ CNY)		Sum (10 ⁶ CNY)	
	Case I	Case VI	Case I	Case VI	Case I	Case VI
I	0.00	3.59	3.13	7.11	3.13	10.70
II	2.39	0.09	3.64	19.15	6.03	19.24
III	0.66	0.38	3.12	22.40	3.78	22.78
IV	1.75	0.30	1.86	18.26	3.61	18.56
Total	4.80	4.36	11.75	66.92	16.55	71.28

4) Handling of multi-objective optimization

The coordinated planning method of FDN is essentially a multi-objective optimization model. In this paper, the multiple interests are formulated and normalized as costs, which can be summed as a single-objective function in a straightforward way. Then, the FDN planning model can be converted into a mixed-integer second-order conic programming (MISOCP) problem, which can be solved effectively by commercial solvers. In this sense, the proposed planning method is to maximise social benefits and achieve the overall economy efficiency for power companies, energy suppliers, and users.

As for multi-objective optimization problems, the Pareto frontier is also known as a promising approach, which provides a wide range of alternative solutions for decision-making. In the Pareto frontier, each objective is considered as equally good. Additionally, game theory is also suited to solving multi-stakeholder problems based on Nash equilibrium, where the interests of power companies, energy suppliers, and users are optimized simultaneously. This paper focuses on how to establish a flexible interconnected architecture of FDN based on SOPs, and the handling of multi-objective optimization will be further studied in future works due to the limited space.

In summary, by establishing a single-objective optimization model, the coordinated planning method of FDN can not only formulate SOP allocation schemes for power companies, but also proactively provide

comprehensive guidance for energy suppliers and users. In this way, the construction of FDNs can cover the interests of multiple parties and offer better power supply services in practical operations.

The detailed modifications are shown as follows.

“

The FDN planning is conducted from the perspective of power companies, who aim to optimize the siting and sizing of SOPs for the distribution network to improve its hosting capability of EVCSs and PVs. The increasing EVCSs and PVs will change the power flow of distribution networks, which affects the location and capacity of SOPs [20]. Thus, the power companies have the motivation to perform a coordinated planning of SOPs, EVCSs and PVs, and the FDN planning model should accommodate the allocation strategy of EVCSs and PVs. On the other hand, as the EVCSs and PVs in distribution networks are invested and built by public stakeholders, the planning strategy of EVCSs and PVs is provided as guidance and suggestion for them. Therefore, the coordinated planning of SOPs, EVCSs and PVs to maximise the overall social benefits is investigated in this paper, which provides comprehensive planning guidance for power companies, energy suppliers and users in distribution networks.

” (Section 1, page 2, highlighted in blue)

“

With the orientation of coordinated planning results, the EVCSs and PVs constructed at the suggested positions will be better regulated and supported by SOPs to ensure the secure hosting of renewable generation and charging load. Therefore, it is beneficial to formulate planning guidance for individual users, especially for the energy suppliers that make profits by providing public generating and charging services. With the simultaneous consideration of grid, source and load in a multi-stage planning model, the planning guidance exhibits high compatibility for all participants in the distribution network. In this way, the construction of distribution network can cover the interests of multiple parties and offer better power supply service in practical operations. At the same time, note that the energy suppliers and users are not mandated to follow the allocation result of sources and loads. But the user-built EVCSs and PVs may bring operational violations to the distribution network, resulting in the inability to plug themselves into the grid. Under such conditions, users have to pay additional costs to dispatch energy storage systems or other controllable resources for regulations, while wasting the public services provided by configured SOPs.

An additional case is investigated as Case VI, where the siting and sizing of SOPs are formulated without the consideration of increasing EVCSs and PVs, while the allocation of EVCSs and PVs is determined and invested by users. Compared with Case I, the number and capacity of SOP in Stage III-IV are smaller, and SOP investment cost is reduced by 9.17%. However, for a stochastically generated allocation of EVCSs and PVs, the voltage violation risk of the distribution network in Stage II-IV exceeds 15%, while the line overload risk exceeds 25%. As a result, the operational cost for violations increases by 4.70 times, and the total cost increases by 3.31 times. The case indicates that if FDN planning is not performed in a coordinated framework, the regulation ability of SOPs cannot be fully utilized, and the randomness caused by sources and loads will exacerbate the operation of distribution network. The separate decision-making leads to worse economic outcomes for the overall benefit of society.

The coordinated planning method of FDN is essentially a multi-objective optimization model. In this paper, the multiple interests are formulated and normalized as costs, which can be summed as a single-objective function in a straightforward way. Then, the FDN planning model can be converted into a mixed-integer second-order conic programming (MISOCP) problem, which can be solved effectively by commercial solvers. In this sense, the proposed planning method is to maximise social benefits and achieve the overall economy efficiency for power companies, energy suppliers, and users. As for multi-objective optimization problems, the Pareto frontier is also known as a promising approach, which provides a wide range of alternative solutions for decision-making. In the Pareto frontier, each objective is considered as equally good. Additionally, game theory is also suited to solving multi-stakeholder problems based on Nash equilibrium, where the interests of power companies, energy suppliers, and users are optimized simultaneously. This paper focuses on how to establish a flexible interconnected architecture of FDN based on SOPs, and the handling of multi-objective optimization will be further studied in future works due to the limited space.

” (Section 8, page 10, highlighted in blue)

Thanks again for your valuable comments!

RESPONSES TO REVIEWER 3'S COMMENTS:

The authors have effectively responded to the issues raised by this reviewer, and I do not have any additional inquiries.

Authors' Response & Revision:

We sincerely appreciate the reviewer's great efforts in reviewing our paper. Thank you again for your valuable comments and encouragements.